# Function and dynamics of the intrinsically disordered carboxyl terminus of β2 adrenergic receptor

Jie Heng[1,2,3,4], Yunfei Hu[5,6], Guillermo Pérez-Hernández [7], Asuka Inoue [8], Jiawei Zhao[4,9], Xiuyan Ma[1], Xiaoou Sun[1], Kouki Kawakami [8], Tatsuya Ikuta [8], Jienv Ding[5,10], Yujie Yang[5], Lujia Zhang[9], Sijia Peng[9], Xiaogang Niu[5], Hongwei Li[5], Ramon Guixà-González [11], Changwen Jin [5], Peter W. Hildebrand [7,12,13], Chunlai Chen [2,3,4,9] ✉ & Brian K. Kobilka [14] ✉

Advances in structural biology have provided important mechanistic insights into signaling by the transmembrane core of G-protein coupled receptors (GPCRs); however, much less is known about intrinsically disordered regions such as the carboxyl terminus (CT), which is highly flexible and not visible in GPCR structures. The β2 adrenergic receptor's (β2AR) 71 amino acid CT is a substrate for GPCR kinases and binds β-arrestins to regulate signaling. Here we show that the β2AR CT directly inhibits basal and agonist-stimulated signaling in cell lines lacking β-arrestins. Combining single-molecule fluorescence resonance energy transfer (FRET), NMR spectroscopy, and molecular dynamics simulations, we reveal that the negatively charged β2AR-CT serves as an autoinhibitory factor via interacting with the positively charged cytoplasmic surface of the receptor to limit access to G-proteins. The stability of this interaction is influenced by agonists and allosteric modulators, emphasizing that the CT plays important role in allosterically regulating GPCR activation.

Allosteric modulation of G-protein coupled receptors (GPCRs) plays a pivotal role in shaping their transmembrane signaling behavior[1]. Prevailing evidence suggests that GPCRs are not simple on/off switches, but exist in an equilibrium of functionally different conformational states[2]. GPCRs, like other signaling proteins[3,4], have intrinsically disordered regions, primarily in the amino terminus, the carboxyl terminus (CT), and the third intracellular loop (ICL3)[5]. The conformational heterogeneity of these regions has been underappreciated as a regulator of GPCR signaling[6]. The CT and ICL3 of GPCRs contain multiple sites for post-translational modification, including sites for phosphorylation and ubiquitinoylation, which modulate interactions with downstream signaling and regulatory proteins[6,7]. Through the

[1]School of Medicine, Tsinghua University, Beijing 100084, China. [2]Beijing Advanced Innovation Center for Structural Biology, Tsinghua University, Beijing 100084, China. [3]Beijing Frontier Research Center for Biological Structure, Tsinghua University, Beijing 100084, China. [4]Tsinghua-Peking Joint Center for Life Sciences, Tsinghua University, Beijing 100084, China. [5]Beijing Nuclear Magnetic Resonance Center, College of Chemistry and Molecular Engineering, Peking University, Beijing 100871, China. [6]Innovation Academy for Precision Measurement Science and Technology, Chinese Academy of Science, Wuhan 430071, China. [7]Charité Universitätsmedizin Berlin, corporate member of Freie Universität Berlin and Humboldt-Universität zu Berlin, Institute of Medical Physics and Biophysics, Charitéplatz 1, 10117 Berlin, Germany. [8]Graduate School of Pharmaceutical Sciences, Tohoku University, Sendai, Miyagi 980-8578, Japan. [9]School of Life Sciences, Tsinghua University, Beijing 100084, China. [10]College of Life Sciences, Peking University, Beijing 100871, China. [11]Condensed Matter Theory Group, Paul Scherrer Institute, CH-5232 Villigen, PSI, Switzerland. [12]Institute of Medical Physics and Biophysics, University Leipzig, 04107 Leipzig, Germany. [13]Berlin Institute of Health, 10178 Berlin, Germany. [14]Department of Molecular and Cellular Physiology, Stanford University School of Medicine, Stanford, CA 94305, USA. ✉e-mail: chunlai@mail.tsinghua.edu.cn; kobilka@stanford.edu

application of structural biology and various biophysical approaches, we have learned much about the structure and dynamics of the 7 transmembranes (TM) core of Family A GPCRs; however, we know relatively little about the intrinsically unstructured segments, which are often not observed in structures determined by crystallography or cryo-electron microscopy.

A recent integrative study highlights the importance of non-transmembrane segments in the modulation of GPCR-mediated signaling and drug responses, which can diversify into distinct signaling properties caused by alternative splicing or expression patterns in different tissues[8]. For instance, sequence divergence resulting from splicing in the receptor CT and ICL3 may alter ligand binding or efficacy, receptor coupling to the downstream effector, or internalization and membrane trafficking. Thus, alternative splicing provides insights into the functional importance of intrinsically disordered regions. Truncation of the intrinsically disordered CT revealed its important role in regulating the constitutive activity of several other GPCRs[9,10]; although different mechanisms were proposed, including interactions with regulatory proteins such as arrestins. For example, two prostaglandin EP3 receptor isoforms differ in constitutive activity revealing that the CT plays a role in maintaining the receptor in its inactive conformation[11]. Truncation of the CT of the avian $\beta_1$AR led to increased activity in cells, but not in purified $\beta_1$AR protein[10]. In the case of the thyrotropin-releasing hormone receptor[12] and the $A_1$-adenosine receptor[13], the enhanced basal activity could be attributed to truncation before the palmitoylation site.

Complementary biophysical approaches have been used to characterize the dynamic behavior of GPCRs[14,15]. Cell-based studies using fluorescence resonance energy transfer (FRET) between two fluorescent probes in the ICL3 and CT show a rapid reduction of FRET efficiency upon agonist stimulation, indicating an increase in distance between the end of the CT and ICL3 upon receptor activation[16–18], possibly due to G-protein engagement. Interestingly, recent nuclear magnetic resonance (NMR) studies found that the phosphorylated CT of the $\beta_2$AR allosterically alters the conformation of core segments by directly interacting with positive charges in the lipid bilayer or the cytoplasmic ends of TM segments. However, these studies did not reveal direct interactions between the unphosphorylated CT and the intracellular surface of the receptor[19].

The $\beta_2$ adrenergic receptor ($\beta_2$AR) has been one of the most extensively studied GPCRs. It primarily signals through Gs but has also been shown to couple to Gi[20], and is one of the first non-visual GPCRs shown to be phosphorylated by several kinases, including PKA[21], PKC[22], and GPCR kinases (GRKs)[23,24], and to couple to $\beta$-arrestins[25]. While much of the intracellular surface of the $\beta_2$AR is disordered, it plays an important role in mediating interactions with these signaling and regulator proteins. The intracellular loop 2 (ICL2) of the $\beta_2$AR has no secondary structure in the inactive state, but transitions into an α-helix in response to receptor activation[26], and is a key determinant of Gs and Gi coupling selectivity[27]. The intracellular loop 3 (ICL3) of the $\beta_2$AR is involved in the engagement of downstream signaling proteins, including G-proteins[28,29], and GRKs[30]. The largest disordered region of the $\beta_2$AR is the CT, which is primarily known for its role in $\beta$-arrestin recruitment[31], receptor internalization[32], and targeting receptors for recycling[33] or lysosomal degradation[34]. Some short linear peptide motifs in the CT, including PDZ binding motif, direct receptors to specific signaling compartments through interactions with scaffolding proteins[35–37]. Post-translational modifications in ICL3 and CT of the $\beta_2$AR, typically phosphorylation[24] and ubiquitination[38] that are frequently observed in intrinsically disordered regions[6,39], play a role in coordinating the cellular signaling dynamics in space and time[40,41].

Here, we report that the CT of several Gs-coupled GPCRs plays an autoinhibitory function in constitutive receptor activation. To explore a potential mechanism of the intrinsically disordered CT in the direct regulation of receptor function, we apply single-molecule fluorescence

resonance energy transfer (smFRET), NMR spectroscopy, and molecular dynamics simulation to monitor the dynamics of the β2AR CT and its interactions with the TM core. We reveal that the β2AR CT regulates basal and agonist-stimulated Gs activation mainly through reversible interactions between negatively charged amino acids in the CT and positively charged amino acids primarily located in ICL2 and ICL3. Furthermore, agonists and a positive allosteric modulator weaken interactions between the 7TM core and the CT through distinct mechanisms. These studies suggest that the CT of some family A GPCRs plays important roles in allosterically regulating signaling beyond their function in desensitization and subcellular localization.

## Results

### Autoinhibitory role of the C-terminus in Gs-coupled GPCRs
It has been suggested that signaling proteins tend to have disordered and autoinhibitory sequences and contain multiple phosphorylation sites and structural variability[4]. As noted above, many GPCRs have a long, unstructured CT. To investigate whether such intrinsically disordered regions would negatively modulate constitutive activation of GPCRs via arrestin-independent manner, we measured cAMP levels in several Gs-coupled GPCRs expressed in HEK293 cells lacking $\beta$-arrestin1 and 2, and $\beta_2$AR (TKO HEK293)[42]. We first examined the $\beta_2$AR, a prototypical Family A GPCR that has a long-disordered CT. The TKO HEK293 cells were transfected with different $\beta_2$AR CT truncations ($\Delta$C) after the palmitoylation site at C341 (Supplementary Fig. 1a). We co-expressed Glo22F cAMP biosensor with receptors to monitor cAMP levels in cells. In parallel, we used a HiBiT-based assay to monitor cell surface expression levels. We found that $\Delta$C346 and $\Delta$C356 show statistically higher constitutive signaling than WT $\beta_2$AR or $\Delta$C378 (Supplementary Fig. 1b,c). To determine if the CT of other Class A GPCRs may also affect the constitutive receptor activation, we selected several Gs-coupled receptors with long CT. Among them, several GPCRs show a similar surface expression-dependent cAMP accumulation as observed for the $\beta_2$AR, including the adenosine subtype 2 A receptor (A2AR), the dopamine subtype 1 receptor (D1R), the dopamine subtype 5 receptor (D5R), and the serotonin subtype 6 receptor (5-HT6R) (Supplementary Fig. 1d). We truncated the CT at 5-7 amino acids away from the palmitoylation site, except for the A2AR that doesn't have a palmitoylation site (Supplementary Table 1). Interestingly, all CT truncated receptors exhibit greater spontaneous receptor activation than WT receptors (Supplementary Fig. 1d, e). It should be noted that the constitutive cAMP activity that we observe for the truncated A2AR is not in agreement with a recently reported study[43]. The difference is likely due to differences in cell surface expression of the truncated A2AR.

We focused on the $\beta_2$AR to investigate the autoinhibitory mechanism of the CT. To monitor ligand-induced acute G-protein activation, we used the NanoBiT G-protein dissociation assay[44]. By using flow cytometry analysis, we first assessed transfection conditions where WT and three $\Delta$CT $\beta_2$AR constructs (all containing N-terminal FLAG tag) expressed at equivalent levels in the TKO HEK293 cells. We matched the expression levels of $\Delta$CTs by reducing the amount of WT plasmid DNA (40 ng or 80 ng) (Fig. 1a). In these expression-matched conditions, removal of the $\beta_2$AR CT leads to a statistically significant enhancement of Gs activation (indicated by increased $E_{max}$ and $pEC_{50}$ values) in response to full agonists (epinephrine and isoproterenol) and partial agonist (clenbuterol) as compared with WT $\beta_2$AR response (Fig. 1b–e, and Supplementary Fig. 2a, b).

To determine if the effect of CT truncation on basal and agonist-stimulated G-protein activity can be observed in purified $\beta_2$AR in detergent, we inserted a sortase A (SrtA) cleavage site at the beginning of the CT ($\beta_2$AR-$s$-CT) (Supplementary Fig. 1a). Purified, $\beta_2$AR-s-CT was preincubated in the absence or presence of SrtA at RT for 2 h to cleave the CT (Supplementary Fig. 2c), then assayed for the ability to stimulate GTP turnover in Gs by measuring residual GTP levels with the

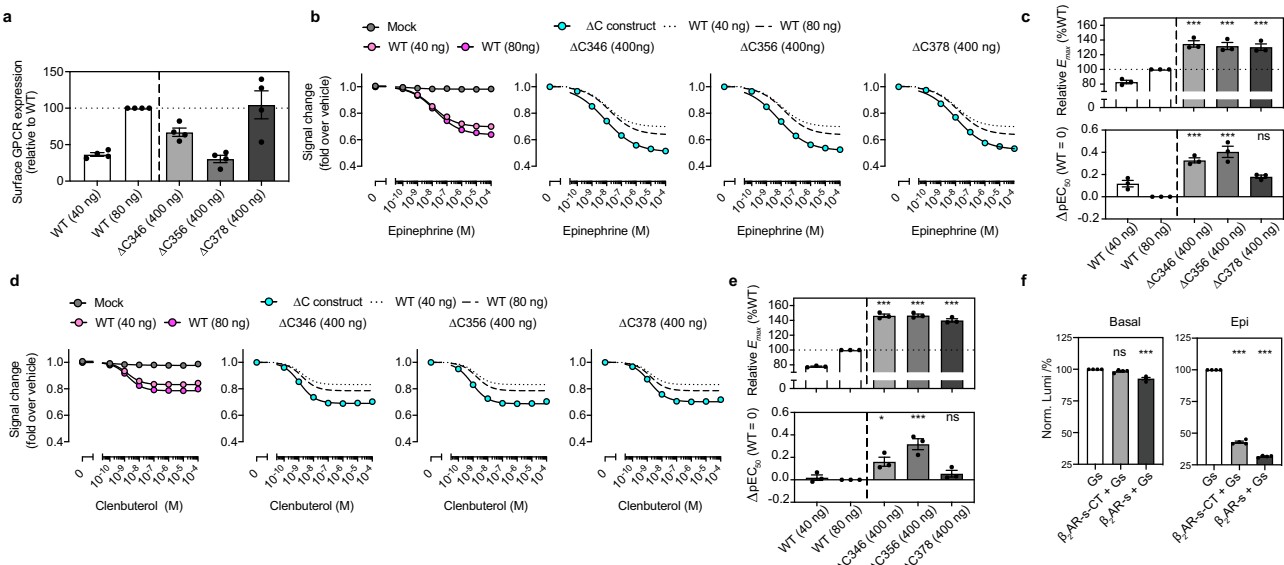

**Fig. 1 | The β2AR CT autoinhibits agonist-stimulated receptor activation. a** Cell surface expression measured by the flow cytometry analysis. Note that amount of plasmid transfection of WT β$_2$AR was titrated down to 40 ng and 80 ng to adjust the expression levels equivalent to the ΔCT β$_2$AR. Data are mean±s.e.m. of 4 independent experiments. **b** Concentration-response curves of epinephrine-induced Gs heterotrimer dissociation in cells. The β$_2$AR-mediated Gs dissociation was measured by the NanoBiT-G-protein dissociation assay, stimulated by full agonist epinephrine in WT and the three ΔCT receptors. Data are mean±s.e.m. of 3 independent experiments. Note that in many data points error bars are smaller than the size of symbols and thus are not visible. **c** The relative $E_{max}$ and $ΔpEC_{50}$ parameters of the Gs dissociation responses stimulated by epinephrine in (**b**). **d** Partial agonist clenbuterol stimulated Gs dissociation in WT and three ΔCT receptors. Data are mean±s.e.m. of 3 independent experiments. **e** The relative $E_{max}$ and $ΔpEC_{50}$ parameters of the Gs dissociation responses stimulated by clenbuterol in (**d**). **f** Removal of the β$_2$AR CT, by pretreating the β$_2$AR-s-CT with/without the SrtA, enhances the GTP turnover in the apo receptor and epinephrine-occupied receptor. Bars and error bars represent the mean±s.e.m. of 4 independent experiments. Statistical significances between ΔCT and expression-matched WT (ΔC346 vs 80 ng; ΔC356 vs 80 ng; ΔC378 vs 40 ng) in **c** and **e** were performed by one-way ANOVA with the Sidak's multiple comparison test. Statistical significances between Gs and the receptor-Gs mixture in **f** were performed by one-way ANOVA with Dunnett's multiple comparison test. $^{***}P < 0.001$; ns, $P > 0.05$.

GTPase-Glo assay. The removal of the β$_2$AR CT enhances both basal and epinephrine-stimulated GTP turnover (Fig. 1f).

## Using single molecule FRET to probe the dynamics of the β$_2$AR C-terminus

The TM6 dynamics of the β$_2$AR have previously been studied by smFRET using a modified β$_2$AR (min-C-β$_2$AR) where all reactive cysteines have been mutated and two reactive cysteines were introduced: N148C at the cytoplasmic end of TM4 as a reference and L266C at the end of TM6[15]. We have previously shown that the ligand binding and in vitro Gs coupling ability of min-C-β$_2$AR and the N148C mutation were preserved[15]. To investigate the conformational dynamics of the β$_2$AR CT, we started with min-C-β$_2$AR and added back two native cysteines in the CT (C378 or C406) together with N148C to generate two constructs: 148C-378C and 148C-406C. These modified receptors were purified from the *Sf9* insect cell and dephosphorylated prior to labeling to minimize receptor heterogeneity arising from phosphorylation during expression in insect cell[24]. The receptor was labeled with Alexa Fluor 555 and Cy5 (Supplementary Fig. 3a, b), immobilized by the amino-terminal Flag epitope to polyethylene glycol (PEG) passivated coverslips (Fig. 2a), and smFRET experiments were conducted on a home-built objective-based total internal reflection fluorescence (TIRF) microscope.

We monitor the FRET efficiency histogram of the unliganded 148C-378C and 148C-406C. Both receptors show a similar single peak distribution, with a FRET center at $0.72 ± 0.01$ (mean of three replicates±s.e.m.) for 148C-378C and $0.56 ± 0.01$ for 148C-406C (Supplementary Fig. 3c). A full-width at half-maximum height (FWHM) value was smaller for the 148C-378C construct ($0.16 ± 0.01$) than for the 148C-406C construct ($0.23 ± 0.02$), which may result from the non-linear dependence of FRET on the distance, where the middle FRET peak is more sensitive to distance changes than low FRET or high

FRET peak. However, we can't rule out the possibility that the middle region of CT is more conformationally homogeneous than the distal CT. To exclude the possibility that the immobilization procedure affects the FRET distribution, we used single-molecule confocal microscopy to verify that the free-diffusing labeled 148C-378C in solution shows a similar FRET distribution centered at 0.73 (Supplementary Fig. 3d).

The β$_2$AR CT was previously observed to be relatively unstructured based on ensemble FRET measurements[45] and NMR chemical shifts[19]. Based on the position of cysteine 378 and 406 in the CT and the Förster radius of our fluorophore pair, we expected smaller FRET values for 148C-378C and 148C-406C if the CT was extended. Given that the CT has a net negative charge and the cytoplasmic surface of the core bundle has a net positive charge (Supplementary Fig. 3e), we investigated the possibility that electrostatic interactions may contribute to interactions between the CT and cytoplasmic surface of the receptor. Therefore, we monitored the FRET distribution for 148C-378C in the presence of different concentrations of NaCl. As can be seen in Supplementary Fig. 3f, with a stepwise titration of NaCl from 50 mM to 1 M, the FRET center decreases from 0.78 to 0.53. The impact of NaCl is reversible with the exchange from high salt to low salt (Supplementary Fig. 3g).

To test the possible interaction between CT and receptor core, we formed a complex with Nb6B9, a G-protein mimic nanobody that stabilizes the active conformation of the β$_2$AR[46]. Nb6B9 indeed causes a leftward shift of the FRET peak in both 148C-378C and 148C-406C (Supplementary Fig. 3h), and the FRET shift exhibits a dose-dependent effect with different concentrations of Nb6B9 (Supplementary Fig. 3i,j). The $EC_{50}$ for the impact of Nb6B9 on FRET was calculated to be $5.5 ± 0.15$ nM for 148C-378C and $4.3 ± 0.15$ nM for 148C-406C (Supplementary Fig. 3k), in agreement with the reported value of 6.4 nM for Nb6B9 binding to agonist occupied β$_2$AR receptor[46].

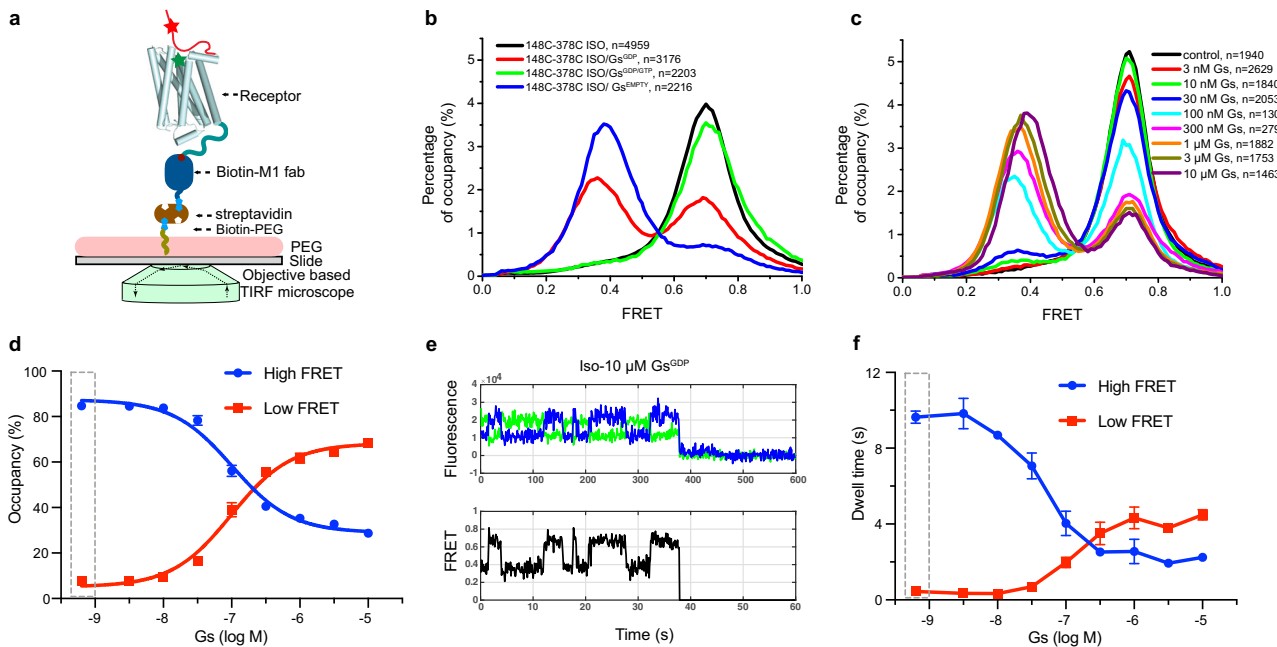

**Fig. 2 | The β2AR CT competes with Gs for binding to the receptor core. a** A schematic model of smFRET experiments on an objective-based TIRF microscope. The receptor core domain is shown as cylindrical helices, unstructured CT as a solid red line, and representative donor and acceptor fluorophores as red and green stars, respectively. **b** Representative FRET histograms of Isoproterenol occupied β2AR (148C-378C) alone, apyrase pretreated Iso-β₂AR-Gs$^{EMPTY}$ ternary complex, Iso-β₂AR-Gs ternary complex with GDP or with a mixture of 30 μM GDP and 300 μM GTP. **c** Representative FRET histograms of the 148C-378C in the presence of different concentrations of Gs$^{GDP}$ with 200 μM isoproterenol. The control sample has isoproterenol alone (black), and the remaining samples include increasing concentrations of Gs$^{GDP}$ from 3 nM to 10 μM. **d** The dose-dependent effect of Gs$^{GDP}$ on the occupancy of high- and low-FRET populations. The cumulative FRET histogram versus Gs$^{GDP}$ concentration in **c** was fitted with a double gaussian model, and the dot

and error bar represent mean±s.e.m. of 3 independent experiments. The dashed rectangular box indicates the control condition without Gs. **e** A representative single-molecule trace of the dynamic ternary complex in the presence of 10 μM Gs$^{GDP}$ and saturating isoproterenol. Donor and acceptor fluorescence intensities collected at 100 ms/frame are shown as green and blue lines, respectively. FRET efficiency is shown as a black line. The sudden fluorescent drop at ~38 s of this trace represents the photobleaching of the donor fluorophore. The smFRET trace reveals transitions between low-FRET and high-FRET states (see also Supplementary Fig. 5h, i). **f** The formation of agonist-β₂AR-Gs ternary complex increases the low-FRET dwell time while decreasing the high-FRET dwell time. The dashed rectangular box indicates the control condition without Gs$^{GDP}$. The high-FRET dwell time at high Gs$^{GDP}$ concentration plateaus at about 2 s. Bars and error bars represent the mean ±s.e.m. of 3 independent experiments.

## The effect of orthosteric and allosteric ligands on β₂AR CT dynamics

Given that CT has effects on basal and agonist-stimulated G-protein activation, we examined the effects of orthosteric ligands with different efficacy profiles on CT dynamics. We observed a similar FRET peak position for the full agonist isoproterenol, inverse agonist occupied receptor and apo receptor (Supplementary Fig. 4a). No statistically significant differences were observed in those three conditions (Supplementary Fig. 4b). A previous study using ensemble FRET showed small but significant ligand-dependent changes in FRET between the fluorophore 4′,5′-bis(1,2,3-dithioarsolan-2-yl)-fluorescein (FlAsH) on the distal CT and Alexa Fluor 568 labeling the native C265 at the cytoplasmic end of TM6[45]; however, this may be due primarily to conformational changes in TM6. Our smFRET measurement performed at 100 ms temporal resolution is not sensitive enough to detect spontaneous fluctuation in traces of unliganded or ligand-occupied receptors (Supplementary Fig. 4c-e). Thus, the single peak suggests that the CT exists primarily in one major conformation or two or more conformations that exchange on a much faster time scale than we can measure.

A number of diverse allosteric ligand binding sites on GPCRs have been revealed by pharmacological and structural studies[1]. For the β₂AR receptor, three allosteric modulators have been described; two of them bind to intracellular pockets[47,48] and one of them to the lipid-facing surface of transmembrane segments (TMs) 3 and 5[49]. The allosteric sites located in the intracellular region of the β₂AR receptor are far away from the orthosteric pocket, and therefore more likely to affect the dynamic properties of the CT and the intracellular surface.

Cmpd-6 is a positive allosteric modulator of the β₂AR receptor that binds to a pocket created by intracellular loop 2 (ICL2) and the cytoplasmic ends of transmembrane segments (TM) 3 and 4 to stabilize ICL2 in an α-helical conformation (Supplementary Fig. 4f)[48]. We observed that increasing concentrations of cmpd-6 shift the FRET center to the left in β₂AR bound to the agonist isoproterenol, but not bound to the inverse agonist carazolol (Supplementary Fig. 4g, h). The binding site of cmpd-6 is close to fluorophore labeling sites 148 C, to exclude the possibility that cmpd-6 exerts a direct effect on 148 C, we generated one additional construct with labeling sites on ICL3 (261 C) and the CT (378 C). The effect of Nb6B9 and cmpd-6 on FRET distributions using this labeling pair were comparable to those using 148C-378C (Supplementary Fig. 4i). In the inactive-state β₂AR, ICL2 is an unstructured loop, while in the active state, ICL2 forms an α-helix which is stabilized by cmpd-6. Therefore, the dose-dependent effect of cmpd-6 is likely due to a change in the conformation of ICL2 that destabilizes interactions with the CT.

In contrast to cmpd-6, cmpd-15 is a negative allosteric modulator of the β₂AR, which binds to a pocket formed by the cytoplasmic ends of the TM1, 2, 6, and 7 as well as ICL1 and helix 8 (Supplementary Fig. 4j)[47]. We find that cmpd-15 alone or together with inverse agonist led to a small rightward shift in the FRET distribution (Supplementary Fig. 4k). We speculate that, by stabilizing the β₂AR in an inactive state, cmpd-15 enhances the association of the CT with the receptor core.

## The effect of Gs on the β₂AR CT dynamics

The structure of the nucleotide-free β₂AR-Gs complex reveals extensive interactions between Gs and the cytoplasmic surface of the β₂AR

including ICL2, and N-terminal and C-terminal end of ICL3 (Supplementary Fig. 5a)[26]. Given the observed effect of the CT on basal and agonist-stimulated Gs activation, we sought to investigate the influence of Gs coupling on the dynamics of the $\beta_2$AR CT. In contrast to the lack of effect of agonists alone, we observed a substantial decrease in mean FRET upon formation of the nucleotide-free agonist-$\beta_2$AR-Gs complex (Fig. 2b). When the agonist-receptor-Gs complex was pretreated with apyrase for 2 h before immobilization and imaging, the $\beta_2$AR shows a low-FRET peak centered at 0.4. A minor high-FRET peak was observed at 0.7, which indicates a minor population that failed to form a stable complex (Fig. 2b). To verify which subunits directly contribute to the CT displacement, we tested the effect of the G$\alpha$ subunit, using the engineered G$\alpha$s subunit (mini-G$\alpha$s) lacking the $\alpha$-helical domain[50] and the dimeric G$\beta\gamma$ subunits on CT dynamics. The mini-G$\alpha$s subunit produced a dose-dependent leftward shift in FRET, while the G$\beta\gamma$ dimer had no effect (Supplementary Fig. 5b, c). This is expected because the selectivity determinants in G-proteins are primarily located in the G$\alpha$ subunit[28]. However, it should be noted that mini-G$\alpha$s was much less efficient at displacing the CT than heterotrimeric Gs, suggesting the G$\beta\gamma$ contributes to coupling efficiency.

The low-FRET state is lower for $\beta_2$AR coupled to Gs compared to $\beta_2$AR bound to Nb6B9 or in the presence of 1 M NaCl (Supplementary Fig. 5d). This may reflect some persistent interaction of the CT with the cytoplasmic surface of $\beta_2$AR bound to Nb6B9 or in the presence of 1 M NaCl; however, we cannot exclude the possibility that the CT may directly interact with Gs resulting in a more extended conformation of the CT. Evidence for this comes from the observation that in the $\beta_2$AR-Gs complex bound to Nb35, the smFRET distribution is the same as in $\beta_2$AR bound to Nb6B9 (Supplementary Fig. 5d). Nb35 binds to the interface of the Ras domain of G$\alpha$s and G$\beta$ where it masks a patch of positively charged amino acids (blue surface, Supplementary Fig. 5e), suggesting that when freed from the $\beta_2$AR core, the CT engages G$\alpha$s in an extended conformation.

Previous studies suggest that the formation of this nucleotide-free complex occurs through at least one GDP-bound intermediate complex[15,51,52]. Therefore, we examined the effect of GDP-bound trimeric Gs on CT dynamics. We observe a larger fraction of high-FRET molecules than in the $\beta_2$AR-Gs$^{EMPTY}$ complex, consistent with a less stable $\beta_2$AR-Gs complex. When incubating the nucleotide-free $\beta_2$AR-Gs complex with a mixture of GDP (30 $\mu$M) and GTP (100 $\mu$M), the low-FRET population shifts back to high-FRET (Fig. 2b). This is consistent with previous single molecule studies on TM6 dynamics that the stability and lifetimes of the agonist-$\beta_2$AR-Gs complex are dramatically decreased in the presence of physiological concentrations nucleotides[15].

### The role of CT dynamics in $\beta_2$AR activation of Gs
The results presented above suggest that the CT must be displaced from the receptor core before Gs can fully engage. It is possible that the CT exists in a dynamic equilibrium between the core engaged and free state, and that Gs can only couple when the CT is not engaged. Alternatively, Gs might form an intermediate complex with the $\beta_2$AR that destabilizes interactions between the $\beta_2$AR and the CT, allowing the active complex to form. To address these possible mechanisms, we sought to monitor the association and dissociation dynamics of the agonist-$\beta_2$AR-Gs ternary complex. First, we examined how GDP and GTP affect the association and dissociation equilibrium of the ternary complex. As expected, the results show that titration of nucleotides in the presence of a saturating concentration of isoproterenol and a fixed concentration of Gs (0.5 $\mu$M) results in a gradual shift of the FRET distribution to the high-FRET state (Supplementary Fig. 5f, g), reflecting disruption of the agonist-$\beta_2$AR-Gs ternary complex[15]. In the preparation of Gs used in these experiments, unbound GDP was removed by size exclusion chromatography, henceforth GDP-bound Gs lacking any excess GDP will be referred to as Gs$^{GDP}$. Therefore, we can titrate

the effect of Gs by adding increasing amounts of Gs$^{GDP}$ to immobilized $\beta_2$AR bound to isoproterenol (Fig. 2c). A noticeable population of low-FRET emerged when the Gs$^{GDP}$ concentration was higher than 10 nM, with an increasing low-FRET population at higher Gs$^{GDP}$ concentrations (Fig. 2c). Interestingly, the two-peak distribution reaches a plateau at a Gs concentration of 1 $\mu$M, which suggests that complex formation and dissociation reaches a steady state. Overall, the Gs$^{GDP}$ exhibits an ability to shift the $\beta_2$AR FRET population with an apparent $EC_{50}$ value of approximately 100 nM (Fig. 2d).

Though no apparent spontaneous fluctuations between high and low-FRET states were observed in the agonist-bounded receptor and the apyrase pretreated nucleotide-free complex (Supplementary Fig. 4c–e and 5h), obvious fluctuations were observed in the presence of Gs$^{GDP}$ (Fig. 2e and Supplementary Fig. 5i). Trace fluctuations were analyzed with the HaMMy software[53] and in-house MatLab code to characterize the dynamic properties of the $\beta_2$AR-Gs ternary complex. Consistent with forming the agonist-receptor-Gs complex, the dwell time of the low-FRET population gradually increases with increasing Gs$^{GDP}$ concentration (Fig. 2f). In contrast, the dwell time of the high-FRET population gradually decreases and reaches a plateau of approximately 2 s (approximately 20-fold the time resolution for imaging) even at a Gs concentration of 10 $\mu$M (Fig. 2f). The plateau of the high-FRET dwell time suggests a factor other than Gs concentration limits the formation rate of the low-FRET agonist-$\beta_2$AR-Gs ternary complex. This could result from spontaneous dissociation between CT and receptor core-domain, thereby allowing access to Gs and transition to the low-FRET state. These spontaneous dissociations may not be discernable in preparations without Gs because they are very short-lived and not observed due to the temporal resolution of our microscope (100 ms). Thus, it appears that Gs can gain access to the $\beta_2$AR only after spontaneous displacement of the CT. If the initial interactions between Gs and the $\beta_2$AR actively initiated CT dissociation, one would not expect a non-zero plateau of the high-FRET state with increasing concentrations of Gs.

### The effect of ligand efficacy on CT dynamics in the presence of Gs$^{GDP}$
The largest structural difference between the inactive and active $\beta_2$AR structures is a 14 Å outward movement of TM6[26]. The efficacy of ligands likely correlates with their ability to stabilize an outward movement of TM6, thereby affecting the rate of formation and stability of the ternary complex (Fig. 3a)[15]. To provide additional support that CT displacement is relevant to G-protein activation, we compared the FRET populations with ligand efficacy determined by a GTP turnover assay for seven different ligands ranging from full agonists to inverse agonists (Fig. 3b, c). We monitored the FRET distributions in the presence of 10 $\mu$M Gs$^{GDP}$ and saturating concentrations of ligands and observed a positive correlation between ligand efficacy in the GTP turnover assay and occupancy of the low-FRET state (Fig. 3b, c). Full agonists epinephrine and formoterol stabilized a comparable low-FRET population, despite the 1000-fold higher affinity of formoterol relative to epinephrine[54]. Partial agonists fenoterol and procaterol show a similar ability to stabilize the low-FRET population. No significant low-FRET population was observed in $\beta_2$AR bound to the inverse agonist carazolol (Fig. 3b). Ligands with higher efficacy also elicit longer low-FRET dwell times and shorter high-FRET dwell times than low efficacy ligands at 10 $\mu$M Gs$^{GDP}$ concentration (Supplementary Fig. 5j).

We also monitored the effect of both positive and negative allosteric modulators in the presence of Gs$^{GDP}$ and saturating isoproterenol. As expected from their efficacy profiles, the presence of cmpd-6 stabilizes the low-FRET state while cmpd-15 stabilizes the high-FRET state, probably by competing for the intracellular surface with the $\alpha$5-helix of G$\alpha$s (Supplementary Fig. 5k). When we titrate Gs$^{GDP}$ in the presence of 20 $\mu$M cmpd-6 and saturating isoproterenol, the

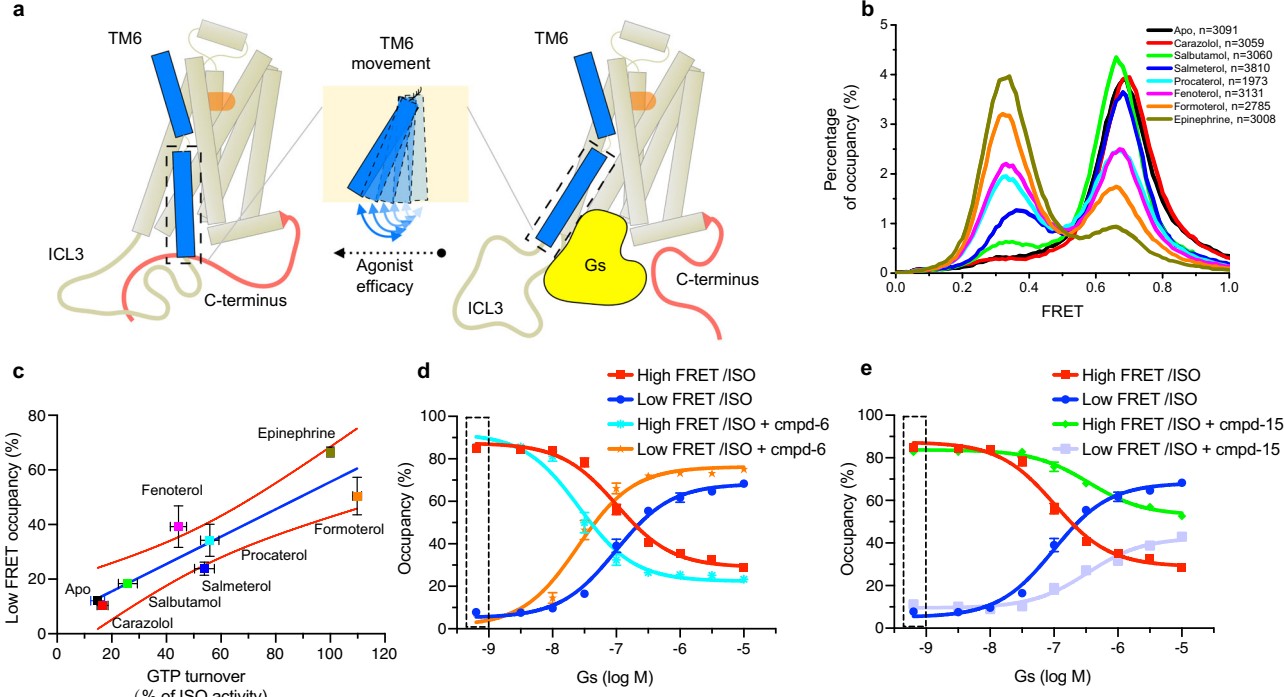

**Fig. 3 | The effect of ligand efficacy and GsGDP on CT dynamics. a** A schematic model outlines the ligand efficacy-dependent influence on the dynamics of TM6. A hallmark of GPCR activation by different agonists is an outward movement of TM6. Gs coupling requires disruption of interactions between the CT and the cytoplasmic core of the $\beta_2$AR. **b** Ligand efficacy-dependent influence on the smFRET efficiency distribution of 148C-378C. All experiments were done in the presence of 10 μM GsGDP, and saturating concentrations of ligands. The stabilization of the low-FRET state correlates with agonist efficacy. **c** The positive correlation between low-FRET occupancy and ligand efficacy (measured by a GTP turnover assay). The statistics of GTP turnover results are mean±s.d. of 3 independent experiments, and the statistics of low-FRET occupancy are mean±s.d. of 2 independent experiments. **d** The concentration titration experiments to evaluate the coupling of GsGDP to the agonist-occupied receptor in the absence or presence of the PAM (cmpd-6). The dashed rectangular box indicates the control condition without GsGDP. **e** The concentration titration experiments to evaluate the coupling ability of GsGDP to the agonist-occupied receptor in the absence or presence of the NAM (cmpd-15). Bars and error bars in (**d**) and (**e**) represent mean±s.e.m. of 3 independent samples.

receptor showed an enhanced ability to couple to GsGDP with a 4-fold lower $EC_{50}$ (26 nM) compared to isoproterenol alone (Fig. 3d). In the presence of 20 μM cmpd-15, the receptor showed an impaired ability to couple to GsGDP with an $EC_{50}$ of approximately 350 nM (Fig. 3e). Interestingly, the dwell time analysis indicates that the cmpd-6 significantly decreases the high-FRET dwell time, while the cmpd-15 prolongs the high-FRET dwell time compared with isoproterenol alone condition (Supplementary Fig. 5l).

**Localizing interactions between the CT and $\beta_2$AR core domain**
In a crystal structure of bovine rhodopsin (PDB ID: 1U19), the C-terminus folds back to directly interact with H8 and the first intracellular loop[55]. In the structure of squid rhodopsin (PDB ID: 2Z73), the C-terminus is an α-helix and forms salt bridges with residues in H8 and ICL3[56]. Nevertheless, bioinformatics indicates that the C-termini of most GPCRs are intrinsically disordered[5,6], and in most crystal and cryo-EM structures of GPCRs the CT is not resolved.

Because of the sensitivity of NMR chemical shifts to local environmental changes[57], measuring chemical shift perturbations is one of the most widely used methods to map protein-protein interfaces. Therefore, we used solution NMR spectroscopy to detect the potential interactions between the CT and core segments. The transpeptidase activity of SrtA has been applied in the segmental labeling of multidomain proteins to help reduce signal overlap in NMR studies[58]. Here, we purified the unlabeled receptor core domain from *Sf9* cells and $^{15}$N-labeled CT from *E. coli*, then ligated the two domains together with SrtA (Fig. 4a). An MBP-fused $^{13}$C/$^{15}$N-labeled CT sample in detergent solution was used to perform triple-resonance NMR experiments, and we assigned the backbone resonances for 57 out of 65

non-proline residues of the CT in the $^1$H-$^{15}$N heteronuclear single quantum coherence (HSQC) spectrum (Supplementary Fig. 6a). The spectra of $\beta_2$AR-[CT] and MBP-[CT] are essentially identical except for slight chemical shift perturbations for a few residues, and the assignments could be directly transferred. The $^1$H, $^{15}$N-HSQC spectrum shows a narrow signal dispersion in the $^1$H dimension, which indicates the CT is intrinsically disordered as previously reported[19]. The effect of CT displacement upon Gs coupling was assessed by comparing both chemical shift changes and signal intensities between the spectra of agonist-bound $\beta_2$AR-[CT] and of agonist-bound $\beta_2$AR-[CT]-Gs treated with apyrase (Supplementary Fig. 6b). The result indicates that the chemical environment of residues in the middle region of the CT, including multiple negatively charged residues between E369 and H390 (Fig. 4b), show greater chemical shift changes than other regions following the formation of a stable $\beta_2$AR-[CT]-Gs ternary complex (Fig. 4c and Supplementary Fig. 6b). Consistent with the hypothesis of CT displacement by Gs coupling, the middle region of the CT shows increased intensity, which may result in faster local dynamics after release from the cytoplasmic core. Interestingly, we found that chemical shift changes of some residues, such as E369, N374, L376, L377, E379, and L388, of the $\beta_2$AR-Gs complex are more similar to the control sample of MBP-[CT] than the $\beta_2$AR-[CT] (Fig. 4d). However, the overall linewidth of $\beta_2$AR-[CT]-Gs is more similar to $\beta_2$AR-[CT] than MBP-[CT]. As shown in Fig. 4d, peaks of K372 and K375 in $\beta_2$AR-[CT] are significantly broadened when compared with those in MBP-[CT], which suggests a conformational heterogeneity for these sites. Above all, the chemical shift changes we observed are likely due to displacement of the CT from receptor core segments, rather than interactions of the CT with Gs, because we observe no

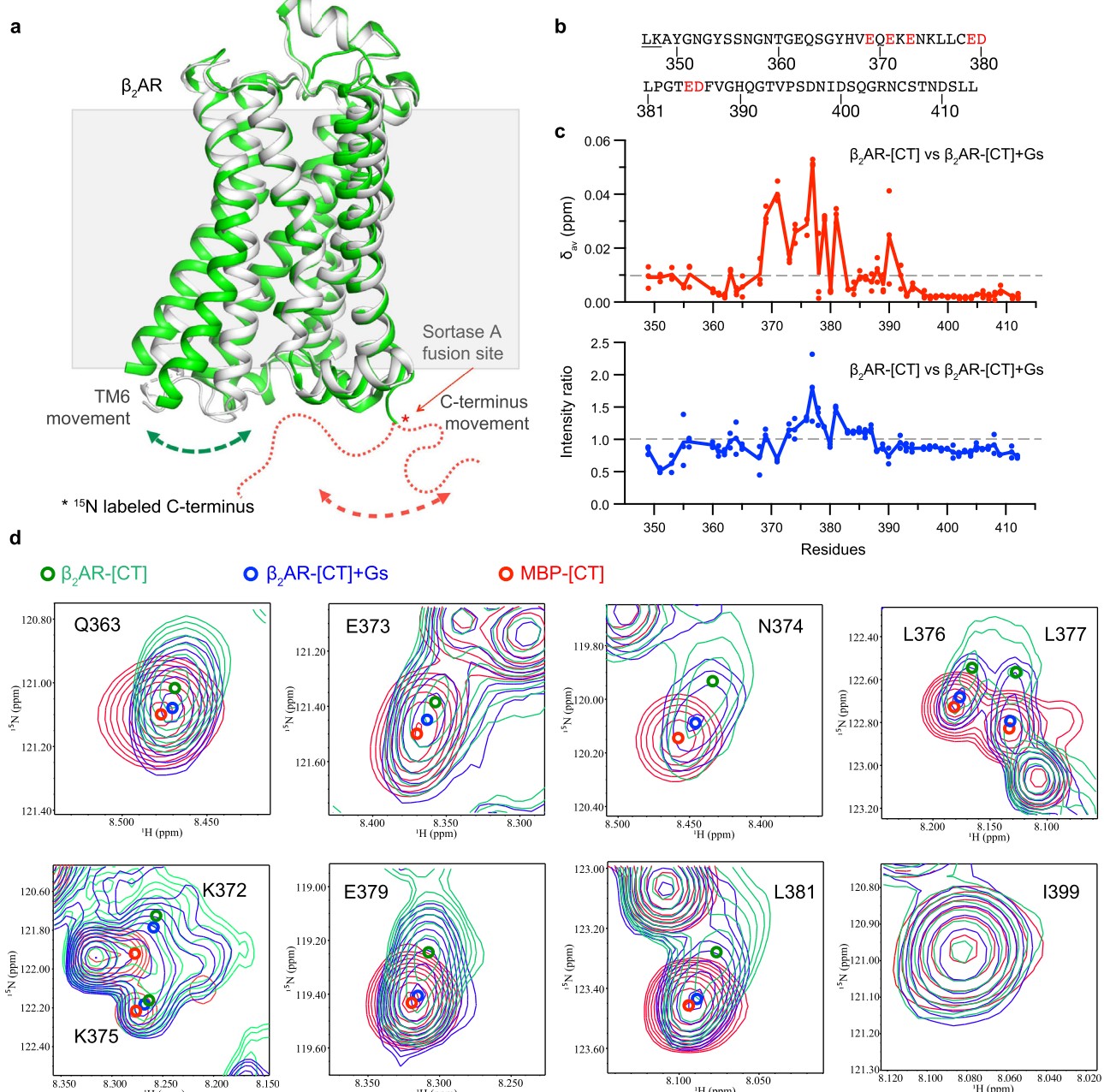

**Fig. 4 | The middle of the β2AR-[CT] interacts with the cytoplasmic surface of the β2AR. a** A schematic model of SrtA ligation of the ¹⁵N isotopic labeled β₂AR-[CT]. The SrtA ligation site was introduced at the end of H8 (see supplementary Fig 1a). The inactive β₂AR structure (PDB ID: 2RH1) is shown in the grey cartoon, while the active structure is shown in the green cartoon (PDB ID: 3POG). **b** The sequence of the β₂AR-[CT] labeled for NMRs studies. The red text highlights the negatively charged residues that show chemical shift changes in panel **c**. **c** The weighted average ¹H–¹⁵N chemical shift changes (Δδ_av) and intensity ratio for each ¹⁵N β₂AR-[CT] residue (347-413) in the absence and presence of Gs protein. Weighted average ¹H–¹⁵N chemical shift changes were calculated as $\Delta\delta_{av} = ((\Delta\delta_H)^2 + (\Delta\delta_N/5)^2)^{1/2}$. The intensity ratio was calculated as $I_{R\text{-}Gs}/I_R$. Dots on the curve indicate two sequential 8 h of measurements of spectra and the merged spectrum. **d** The spectra of agonist-β₂AR-[CT]-Gs complex is more similar to the spectra of MBP-[CT] than the β₂AR-[CT]. Representative peaks from superposed ¹H–¹⁵N HSQC spectra of ¹⁵N MBP-[CT], BI occupied ¹⁵N β₂AR-[CT], and BI occupied ¹⁵N β₂AR-[CT] coupled to nucleotide-free Gs. The peak centers are shown as colored circles.

significant chemical shift perturbation when we mix 50 μM MBP-[CT] with 50 μM Gs (Supplementary Fig. 6c).

Of interest, the segment of the β₂AR CT with the largest chemical shift perturbations contains four acidic residues. As noted above, the sensitivity to salt concentration (Supplementary Fig. 3f) implies that electrostatic interactions between the CT and core segments may play a role in CT engagement.

To evaluate the contribution of 11 negatively charged residues in the β₂AR CT, we grouped them into three clusters, ED1, ED2, and ED3,

and then selectively mutated those clusters to alanine, including combined mutants ED12 and ED123 (Supplementary Fig. 7a). Briefly, smFRET studies show that ED1 has the largest change in the FRET distribution with a shift similar to that observed for ED123, while ED2 has a moderate effect, and ED3 the smallest effect (Supplementary Fig. 7b). Notably, all selected mutants show a further low-FRET shift in the presence of 500 mM sodium chloride buffer or 1 μM Nb6B9 (Supplementary Fig. 7c, d). Thus, mutation of all negatively charged residues does not completely disrupt interactions between the CT and

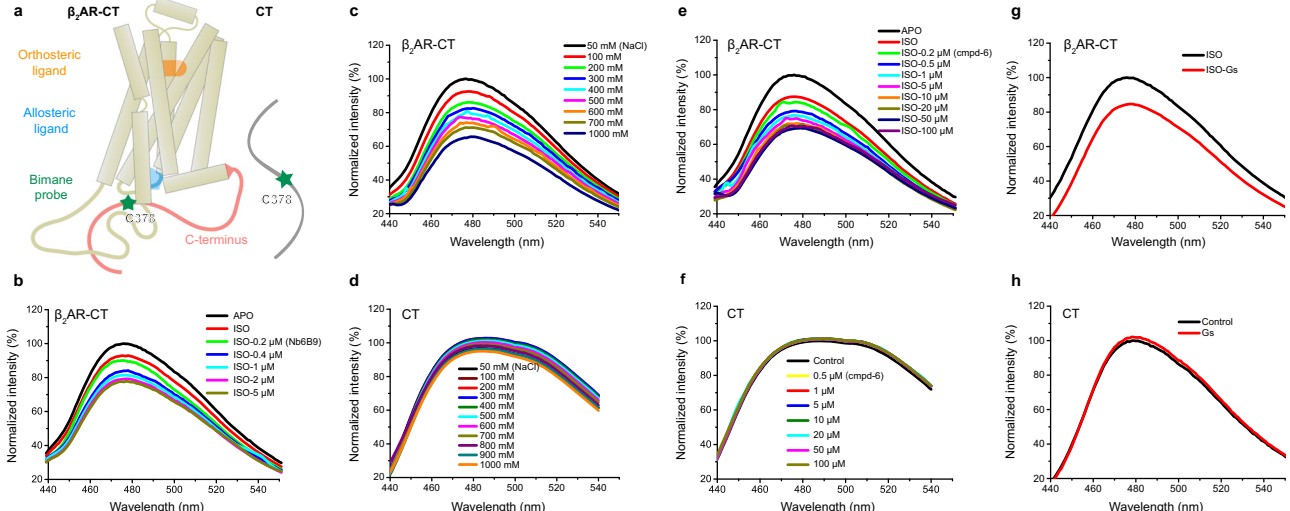

**Fig. 5 | Monitoring interactions between the CT and the cytoplasmic surface with a fluorescent probe bound to C378. a** Model of bimane labeled β2AR-CT and CT at C378. The orthosteric ligand used in this experiment is isoproterenol, and the allosteric ligand is cmpd-6. **b** Increasing concentrations of Nb6B9 in combination with saturating isoproterenol result in a decrease in the bimane intensity of the β2AR-CT, suggesting that the fluorophore moves to a more polar environment. **c** Increasing sodium chloride concentrations decrease bimane intensity in the apo β2AR-CT. **d** The effect of sodium chloride concentration on bimane-labeled CT. **e** Increasing concentrations of the cmpd-6 combined with saturating isoproterenol result in a decrease in the bimane intensity of the β2AR-CT. **f** The effect of cmpd-6 concentrations on bimane-labeled CT. **g** Nucleotide-free Gs reduces the bimane intensity of the β2AR-CT. **h** Gs doesn't affect the intensity of bimane-labeled CT.

TM core segments, suggesting the existence of other interactive determinants, possibly electrostatics involving the few positively charged amino acids in the CT. To investigate the effect of mutagenesis of negatively charged residues on Gs coupling, we analyze ternary complex dynamics in the presence of the same amount of Gs$^{GDP}$. We observed that ED1 and ED123 enhance the low-FRET population while ED2 and ED3 alone had little effect (Supplementary Fig. 7e), suggesting a better Gs coupling efficiency in ED1 and ED123. Since we observed enhanced basal cAMP production and Gs dissociation in CT truncated β2AR above, we sought to evaluate the contribution of negatively charged residues on basal cAMP accumulation and agonist-stimulated Gs dissociation in cells. Both ED1 and ED12 show a higher constitutive activation than the WT receptor (Supplementary Fig. 7f, g). Consistent with this, ED1 and ED12 also had significantly enhanced isoproterenol-stimulated Gs-dissociation responses compared to the WT receptor (Supplementary Fig. 7h–k).

Taken together, the NMR studies and the mutagenesis experiments discussed above suggest that the CT segment between 370 and 380 plays a prominent role in interactions of the CT with the receptor core resulting in the inhibition of basal and agonist-stimulated activation of Gs. To further verify the importance of this region in the middle of the CT, we purified and labeled C378 with the environmentally sensitive fluorophore monobromobimane (bimane), then ligated it to the receptor core (Fig. 5a). The emission intensity of bimane and the wavelength where the maximal emission intensity is observed (λ$_{max}$) are sensitive to the polarity of bimane's chemical environment. We monitored the emission spectrum of bimane before and after the addition of Nb6B9 (Fig. 5b) and observed a decrease in bimane intensity with increasing concentrations of Nb6B9, consistent with bimane moving to a more polar environment. As noted above, increasing concentrations of NaCl are expected to disrupt electrostatic interactions between the CT and receptor core (Supplementary Fig. 3f, g). In agreement with this, we observe a decrease in bimane fluorescence with increasing concentrations of NaCl (Fig. 5c). This is not due to a direct effect of NaCl on bimane, as NaCl has very little effect on the free CT labeled with bimane (Fig. 5d). We also observe a decrease in bimane fluorescence

in the presence of increasing concentrations of cmpd-6 and β2AR coupled to nucleotide-free Gs (Fig. 5e–h).

## The CT mainly interacts with intracellular loops 2 and 3

To further investigate the potential interaction patterns between the CT and the receptor core with atomic-level detail, we carried out molecular dynamics (MD) simulations. We used previously solved 3D structures, and various modeling techniques to build an initial atomistic model of the inactive β2AR comprising amino acids 26-390. In this model, the CT is largely unstructured. We embedded this model in a lipid bilayer solvated by excess water at a physiological salt concentration and simulated the resulting system using an adaptive sampling scheme (see Methods).

Consistent with our NMR data, the CT appears primarily disordered within the timescale of the simulations. These trajectories emerge as an ensemble of states that are conformationally different, short-lived, and quickly interconverting with one another. The two fluorophore labeling sites, 148 C and 378 C, sample a wide range of distances during the simulations (Fig. 6a–c). Although the MD trajectories sample a much shorter timescale (nanoseconds to microseconds) than the FRET experiment (milliseconds), these states display Cys-Cys distances also compatible with the high-FRET value, where the Cys-Cys distance distribution peaked at around 40 Å and very few extend beyond 50 Å (Fig. 6a, the panel on the top right). Hence, the ensemble of CT conformations as a whole can be assumed to be representative of the high-FRET state. Notably, in most of these conformations, the CT tends to adopt non-extended conformations and stay near the receptor by both self-interactions and forming multiple contacts with the receptor core (Fig. 7a and Supplementary Fig.8a–c). As shown in Fig. 7b, Supplementary Fig. 8d–f and Supplementary Table 2, the CT mostly interacts with itself and with the ICL1, ICL2, TM5-ICL3-TM6, and H8 regions of the receptor. While no single interaction appears to dominate, Q337$^{8.55}$-L342$^{CT}$ always tethers the proximal segment of CT to H8. Simultaneously, negatively charged CT residues from ED1 and ED2 clusters interact preferably with positively charged residues from the ICL3-TM6 region of the receptor core (Fig. 7b and Supplementary Table 3). This is also consistent with proximal and

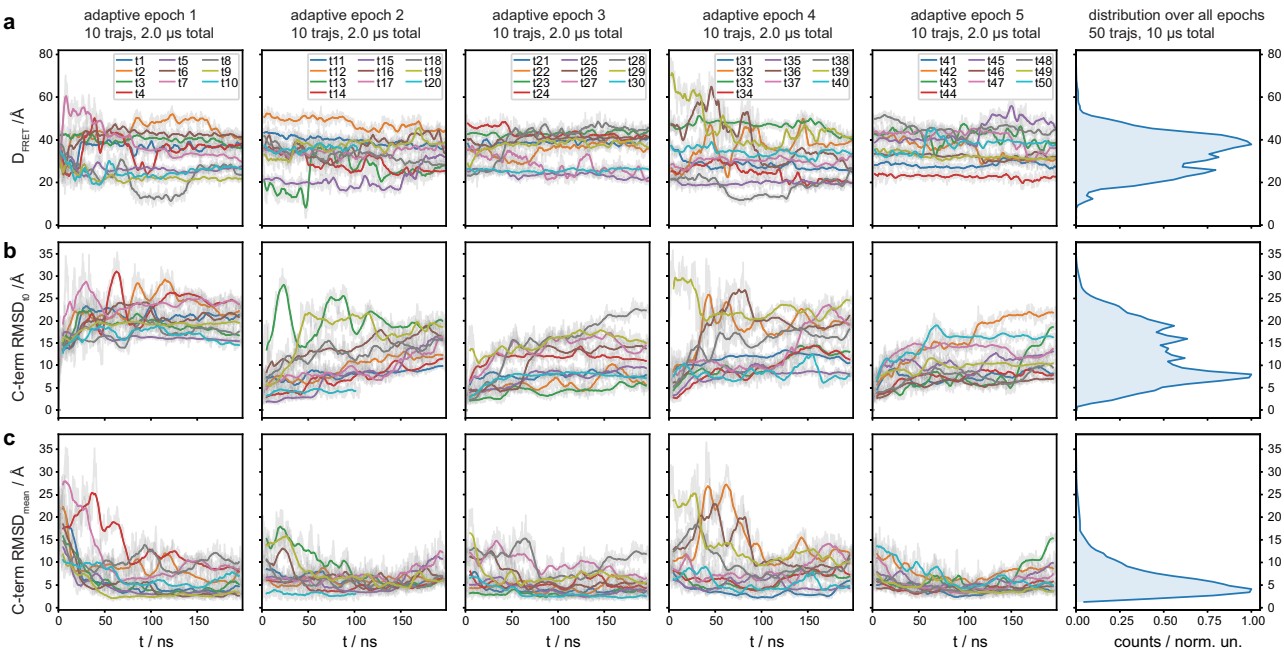

**Fig. 6 | The flexibility of the β2AR CT observed in MD simulations. a−c** The flexibility of the $\beta_2$AR CT observed in MD simulations and the related histogram of all the values of the CT fluctuations around each trajectory's average structures. The first five columns represent the five adaptive sampling epochs, each epoch consisting of ten individual, unbiased, 200 ns long MD trajectories of the $\beta_2$AR. Each row shows the time traces for different geometrical parameters. **a** Minimum distance between CYS148 and CYS378 heavy-atoms, tracking the distance between dyes in the smFRET experiments. This parameter was used as the "exploit" component in the adaptive sampling. We observe a high range of distances sampled in all 50 trajectories, indicating a high structural flexibility of CT in the nanosecond timescale: Only in a few trajectories does the CYS-CYS distance remain constant (e.g. the red curve in the last epoch, trajectory 44). In most cases, it changes by tens

of Angstrom (e.g. the green curve of the second epoch, trajectory 13). **b** Root-mean-square-deviation (RMSD) of the CT backbone atoms, monitoring the structural changes of the CT with respect to each trajectory's starting frame. Strikingly, The CT moves away quickly from the modeled starting conformation by up to 30 Å in the first epoch. The disordered nature of CT is reflected by the high variability of different conformations sampled in all epochs. **c** RMSD of the CT backbone atoms, monitoring the fluctuations of the CT around the trajectory's average structure. The wide range of values hints at a rather flat free-energy landscape underlying the dynamics, which is characteristic of structurally disordered proteins. Time-traces show the running-averages as solid lines (smoothing window of 50 ns) overlaid on top of the raw time-trace, shown in gray in the background. The distributions on the rightmost panels are computed using the raw data.

middle regions of the CT mediating the interaction with the receptor core, as shown by our NMR data.

Further analysis of the MD data shows that the CT receptor contacts change as the distance between the two fluorophore labeling sites, 148 C and 378 C, changes. Contacts with regions close to the membrane, such as ICL1 or ICL2, occur only in a short distance range (148C-378C < 35 Å), whereas contacts with ICL3, which is further away from the membrane, occur predominantly at larger distances (148C-378C 35-60 Å) (Supplementary Fig. 9 and Table 4). Only at extreme distances (148C-378C > 60 Å), which rarely occur in our simulations, are the contacts reduced to the proximal CT and H8, as seen in the structure of the $\beta_2$AR-Nb80 complex[59].

To further validate that the CT indeed interacts with ICL3 as MD simulations suggest, we carried out a paramagnetic relaxation enhancement (PRE) experiment[60], in which we site-specifically attach a spin label that could enhance the transverse relaxation rates of nearby nuclei and thus lead to line-broadening effects. A spin probe, 4-(2-Iodoacetamido)-TEMPOL, was chosen because of the high stability of its unpaired electron and exceptional sensitivity. TEMPOL was attached to S261C at the C-terminal end of ICL3, and the $^{15}$N backbone labeled CT was ligated to the core segment as in the NMR study above. We monitored the PRE effect in the absence and presence of Nb6B9, to mimic the CT displacement upon G-protein coupling. As illustrated in Fig. 7c and Supplementary Fig. 6d, the proximal and middle regions of the CT (sequence before residue 392) are close to the radical center of 261 C in ICL3. The binding of Nb6B9 enlarges the overall distance between the CT and the radical center and, as a result, reduces the PRE effect.

In agreement with the MD and PRE data, removal of the positively charged region by truncation of ICL3 (residues 236-263) leads to a

leftward shift of the high-FRET center (Fig. 7d), similar to the consequence of removing negatively charged residues ED123 in the CT. Neutralization of the four positively charged residues (K140, K147, K149, and R151) to Gln in ICL2 gave a slightly smaller effect on the high-FRET distribution compared to the truncation of ICL3 or removal of negative charges in the CT (ED123). Interestingly, when deletion of ICL3 or ED123 mutations is combined with cmpd-6, the high-FRET center is in a similar position observed for Nb6B9 or 1 M NaCl (Fig. 7e). The cooperative effect suggests that a complex network of interactions between ICL2, ICL3, and the CT is responsible for the observed interactions between the CT and the TM core.

## Discussion

Taken together, using five Gs-coupled receptors with long C-terminus, including $\beta_2$AR, A2AR, D1R, D5R, and 5-HT6R, we demonstrate that the intrinsically disordered CT can play a role in modulating GPCR signaling. Like other intrinsically disordered autoinhibitory domains in other signaling proteins[61], the $\beta_2$AR CT suppresses basal and agonist-stimulated activation of Gs, and this is due to dynamic interactions between the CT and the intracellular surface of the TM core. Our smFRET and NMR experiments show that the CT is predominantly engaged with the cytoplasmic surface (ICL2, ICL3, and H8), primarily through electrostatic interactions that can be disrupted by either sodium chloride, G-protein mimic nanobody, or Gs protein. Dwell time experiments suggest that G-protein access to the activated receptor takes place after spontaneous dissociation of the CT from the $\beta_2$AR core (Fig. 8). The stability of the core interaction, reflected in the high-FRET dwell time, is reduced in the presence of agonists and a positive allosteric modulator.

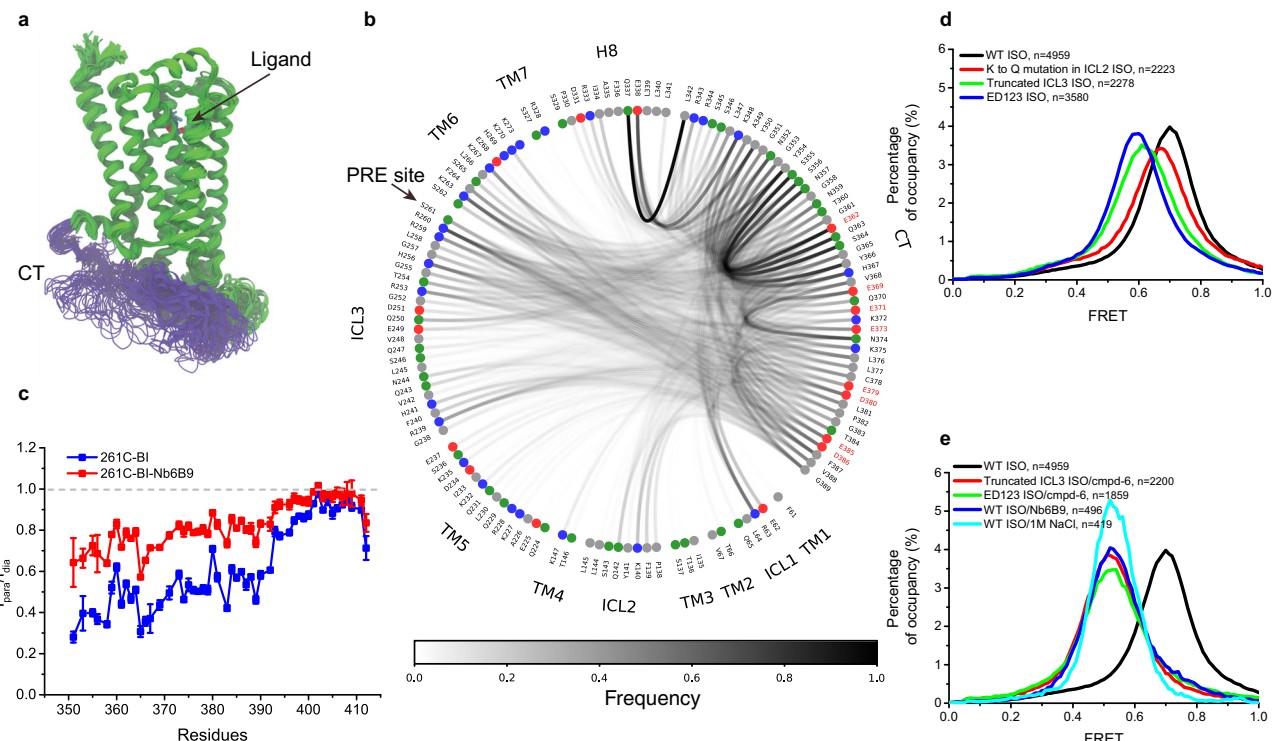

**Fig. 7 | Contacts between the β2AR-[CT] and intracellular loop regions of the β2AR. a** Superimposed snapshots of the β2AR from one representative MD trajectory (out of 50). The CT is shown in purple, and the remainder of the receptor is in green. The green highly mobile region belongs to the ICL3. We show here one snapshot every 2 ns with a smoothing window of five frames from a 200 ns trajectory. **b** Residue-residue contacts across the aggregated 50 MD trajectories (10 μs total simulation). Red and blue dots represent negatively and positively charged residues, respectively, while green and gray ones are neutral hydrophilic and neutral hydrophobic residues, respectively. The curve between two dots indicates that the two respective residues are in contact distance of at least 3.5 Å, with the opacity of each curve representing contact frequency. The negatively charged residues of the CT are further highlighted in red text labels. Only contacts involving the CT are shown. **c** The potential interaction between CT and ICL3 was measured by a paramagnetic relaxation enhancement experiment (PRE). A spin probe IA-TEMPO was attached to residue 261 C at the C-terminal end of ICL3, and the distance-dependent PRE effect was evaluated by the intensity ratio ($I_{para}/I_{dia}$) of $^{1}$H, $^{15}$N-HSQC cross-peaks versus primary sequence at 293 K. The error bars for $I_{para}$ and $I_{dia}$ represent mean and s.e.m. of 3 sequential 8 h measurements. **d** Mutagenesis to remove all the negatively charged residues in the CT (ED123), truncation of ICL3 region 236-263 (Truncated ICL3), and neutralization of the positive charge by Lys to Gln mutagenesis in ICL2 (K to Q mutation in ICL2) affect the FRET distribution. **e** Conditions that show complete disruption of CT and core-domain interactions revealed by FRET distributions of 148C-378C.

From an energy landscape perspective, intrinsically disordered segments of a protein are characterized by a continuum of conformational states and transitions within many flat local minima[3]. Indeed, although we observed long-lived high-FRET states in the smFRET experiments, our MD simulations suggest that the CT does not adopt a single conformation, but rather remains disordered and forms multiple weak and transient interactions with the cytoplasmic surface including ICL2, ICL3, and H8. This ensemble of states is likely represented by the single high-FRET state captured by smFRET due to the much lower temporal resolution compared to the MD simulations. Based on our results, we propose the mechanistic model illustrated in Fig. 8. In the absence of a ligand, engagement of the CT with the cytoplasmic surface prevents access to Gs. Spontaneous dissociations of the CT in unliganded β2AR are infrequent as revealed by long high-FRET dwell times. The binding of agonists or cmpd-6 causes a weakening of interactions between the CT and the cytoplasmic surface, as evidenced by a reduction in the high-FRET dwell time, resulting in enhanced accessibility to Gs.

The physiologic importance of the auto-inhibitory effects of the β2AR CT has yet to be determined; however, we speculate that scaffolding proteins that bind to the CT, such as AKAP7[62], may disrupt the interaction between the CT and the cytoplasmic surface and facilitate coupling to Gs. It is possible that arrestin binding to the β2AR CT may facilitate activation of Gs in an endosomal compartment as suggested by the "megaplex" complex of β2AR-Gs-arrestin[63]. Finally, when the

β2AR is bound to Gs, the more extended CT may be more available for kinases and ubiquitinases.

## Methods

### Cell lines

The human β2AR receptor was expressed in *Spodoptera frugiperda (Sf9)* cells infected with recombinant baculovirus (BestBac, Expression Systems). Human Gs heterotrimer was expressed in *Trichoplusia ni* insect cells with recombinant baculovirus (Bac-to-Bac, Invitrogen). For cAMP Glosensor assay and NanoBiT-G-protein dissociation assay, HEK293 cells deficient for β2AR and β-arrestin1/2[42] were grown in a humidified 37 C incubator with 5% $CO_2$ using Opti-MEM I Reduced Serum Medium (Thermo Fisher Scientific). *E. coli* BL21(DE3) cells were used to express GST-CT fusion protein, mini-Gas, Nb6B9, and SrtA enzyme.

### Plasmids

For constitutive Gs signaling measurement, full-length human GPCRs (β2AR, A2AR, D1R, D5R, 5-HT6R; residues from the second codon to the last codon) and C-terminally truncated GPCRs (2-346, 2-356, 2-378 for β2AR; 2-316 for A2AR; 2-358 for D1R; 2-386 for D5R; 2-347 for 5-HT6R) were inserted into the pCAGGS expression plasmid with the N-terminal haemagglutinin signal sequence followed by the FLAG epitope and the HiBiT tag flanked by with flexible linkers (MKTIIAL-SYIFCLVFA-DYKDDDDK-GGSGGGGSGGSSSGGG-VSGWRLFKKIS-GGSG GGGSGGSSSG). The nucleotide sequence of LgBiT[64] was synthesized

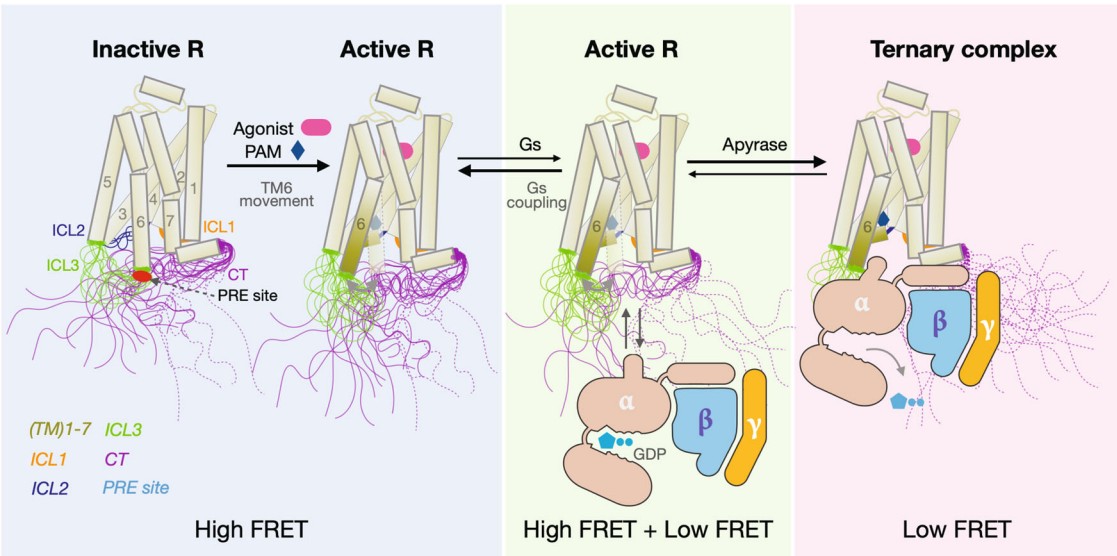

**Fig. 8 | Model illustrating the role of the β2AR-[CT] dynamics in G-protein activation.** In the inactive state, the disordered CT of the β₂AR adopts multiple conformations and directly interacts with intracellular loops of the TM core. The solid lines represent conformations of the CT directly interacting with the TM core, while the dashed lines represent conformations of the CT that have spontaneously dissociated from the TM core. The agonist and positive allosteric modulator promote the outward movement of TM6 and the transition of ICL2 to a helix. These changes result in weakened interactions between the CT and the cytoplasmic surface and facilitate spontaneous dissociation thereby decreasing the high-FRET dwell time and increasing the accessibility of receptors to Gs. Coupling with Gs prevents interactions between the CT and the TM core, stabilizing the low-FRET state. The stability of the β₂AR-Gs complex is dependent on the concentration of GDP and GTP in cells. Apyrase treatment hydrolyses GDP and GTP resulting in the formation of a stable nucleotide-free complex.

after codon optimization for expression in *E. coli* (GenScript) and subcloned into the pET-28a (+) vector at the BamHI-XhoI cloning sites with an N-terminal flexible linker (GGGGSGGGGS). The resulting LgBiT construct contained the pET-28 (+)-derived N-terminal hexahistidine tag (M-GGS-HHHHHH-SSG), thrombin cleavage site (LVPRGS), and T7 tag (H-MASMTGGQQMG-RGS) before the linker-LgBiT. The full-length human β₂AR and three different truncated variants (at 346, 356, and 378) were also cloned into pcDNA3.1 vector with the cleavable haemagglutinin signal sequence followed by the M1 Flag epitope (DYKDDDDA) for NanoBiT assay. For fluorescent labeling at C265, a modified β₂AR (min-C-β₂AR) where all reactive cysteines except C265 have been mutated (C77V, C327S, C341L, C378A, and C406A) was cloned into pVL1392 vector to express in *Sf*9 insect cells and purified with N-terminal FLAG tag as previously described[15]. Two mutations M96T, and M98T that have been reported to increase the receptor expression level were introduced to facilitate NMR and PRE experiments, which need a large amount of receptor[65]. To evaluate the functionality of receptor CT by in vitro assays, a chimera receptor (β₂AR-s-CT) was designed by inserting the SrtA cleavage site (LPETG) at residue 346. For smFRET fluorescent labeling, the mutation N148C or S261C was introduced on the background of min-C-β₂AR with native C378 or C406, respectively. Both solution NMR and PRE experiments were done with the receptor with minimal cysteine background and SrtA ligation sites (min-C-β₂AR-s-1D4 or min-C-β₂AR-261C-s-1D4). For ¹⁵N isotopic labeling of the receptor CT, residues 347-413 followed by a thrombin enzyme site and hexahistidine tag were cloned to the pGEX 6p-1 vector to express and purify from the *E. coli* cells. A TEV protease cleavage site (ENLYFQ/G) was added to the N-terminal of CT to produce a glycine end that is suitable for SrtA fusion after TEV cleavage. The mini-Gαs[50], trimeric Gs[26], and Nb6B9[46] constructs were produced as previously reported. All mutants were generated with site-directed mutagenesis and confirmed by sequencing.

### Preparation of LgBiT
BL21 (DE3) cells were transformed with the LgBiT plasmid and cultured in the LB media at 37 °C until OD600 reached 0.6. At this bacterial cell density, IPTG was added at a final concentration of 1 mM and the cells were further cultured for 3 h. Cells were collected by centrifugation (5000 × *g*, 10 min, 4 °C) and their pellet was then frozen with liquid nitrogen and stored at −80 °C until the purification.

The cells were suspended in buffer A (50 mM Tris, pH 8.0, 100 mM NaCl, 10 mM MgCl₂; typically, 30 ml buffer A for 500 ml LB culture media) and homogenized by a tip ultrasonicator (UD-211, TOMY). The supernatant after centrifugation (17,000×*g*, 30 min, 4 °C) was supplied into Ni-NTA resin (Qiagen) pre-equilibrated with buffer B (50 mM Tris, pH 8.0, 100 mM NaCl, 10 mM MgCl₂, 20 mM imidazole). After incubation for 30 min at 4 °C, the resin was washed with buffer B and eluted with buffer B containing 200 mM imidazole. The peak fractions of the eluate were collected, and an equal volume of 50% (v/v) glycerol was added and stored at −80 °C until use. The activity of the purified LgBiT was measured by mixing the synthesized HiBiT peptide (VSGWRLFKKIS) (GenScript) and furimazine (Chemspace) at concentrations of 1 μM and 10 μM, respectively with a luminescent microplate reader (SpectraMax L, Molecular Devices). The volume of the purified LgBiT solution was adjusted to show the equivalent activity to LgBiT purchased from Promega.

### Constitutive cAMP Glosensor assay
Constitutive Gs signaling by a test GPCR was measured by an in-house modified Glosensor cAMP assay (Promega) and normalized by HiBiT-based surface expression analysis performed in parallel. HEK293 cells deficient for β₂AR and β-arrestin1/2[42] in the growth phase were harvested and suspended in Opti-MEM I Reduced Serum Medium (Thermo Fisher Scientific) at a cell concentration of 4 × 10⁵ cells ml⁻¹, seeded in a 96-well white culture plate (80 μL per well) and placed in a CO₂ incubator. Transfection solution (per well in the 96-well plate hereafter) was prepared by mixing 40 ng of a Glo-22F cAMP biosensor (gene synthesized with codon optimization by GenScript)-encoding pCAGGS plasmid and titrated volumes of the N-terminally HiBiT-tagged GPCR plasmid (from 0.2 ng to 8 ng; 2-fold or 2.5-fold titration) plus a balance of the empty pCAGGS plasmid (total plasmid volume of 28 ng), along with 0.2 μL of 1 mg ml⁻¹ PEI and 20 μL Opti-MEM I

Reduced Serum Medium. Transfection was performed on the same day as cell seeding and the cells were cultured for 1 day. For the Glosensor-based cellular cAMP measurement, 20 μL of the conditioned media were removed and the cells were mixed with 20 μL of 12 mM D-luciferin potassium solution (FujiFilm Wako Pure Chemical) diluted in HBSS containing 0.01% BSA and 5 mM HEPES (pH 7.4) (assay buffer). For the HiBiT-based surface GPCR expression measurement, 20 μL of the conditioned media were removed and the cells were mixed with 20 μL of LgBiT (1:200 of the stock solution) and 50 μM furimazine diluted in the assay buffer. After 2 h (Glosensor) or 30 min (HiBiT) incubation in the dark at room temperature, the luminescence of each well was measured by a microplate luminometer with an integration time of 0.4 s per well with 5 rounds of readings (Spectramax L, Molecular Devices). The luminescent counts were normalized to that of mock-transfected cells prepared in the same plate and expressed as a fold-change value. For individual GPCR constructs, surface expression (HiBiT signal) and cAMP level (Glosensor signal) were plotted and those in linear correlation were used to calculate a slope (expression-normalized cAMP level). For each experiment performed in parallel, a slope value of the C-terminally truncated construct was normalized to that of the full-length construct, and %WT values were shown.

## The cell-surface expression of receptors by flow cytometry analysis

Transfection was performed following the same procedure as described in the "NanoBiT-G-protein dissociation assay" section except for the use of a 6-well culture plate (2 mL culture media and a half volume of the transfection mixture). One day after transfection, the cells were collected by adding 200 μL of 0.53 mM EDTA-containing Dulbecco's PBS (D-PBS), followed by 200 μL of 5 mM HEPES (pH 7.4)-containing Hank's Balanced Salt Solution (HBSS). The cell suspension was transferred to a 96-well V-bottom plate in duplicate and fluorescently labeled with an anti-FLAG epitope (DYKDDDDK) tag monoclonal antibody (Clone 1E6, FujiFilm Wako Pure Chemicals; 10 μg ml$^{-1}$ diluted in 2% goat serum- and 2 mM EDTA-containing D-PBS (blocking buffer)) and a goat anti-mouse IgG secondary antibody conjugated with Alexa Fluor 647 (Thermo Fisher Scientific, 10 μg ml$^{-1}$ diluted in the blocking buffer). After washing with D-PBS, the cells were resuspended in 200 μL of 2 mM EDTA-containing-D-PBS and filtered through a 40 μm filter. The fluorescent intensity of single cells was quantified by an EC800 flow cytometer equipped with dual 488 nm and 642 nm lasers (Sony). The fluorescent signal derived from Alexa Fluor 647 was recorded in an FL3 channel, and the flow cytometry data were analyzed with the FlowJo software 10.8.1 (FlowJo). Live cells were gated with a forward scatter (FS-Peak-Lin) cutoff at the 390 settings, with a gain value of 1.7. Values of mean fluorescence intensity (MFI) from approximately 20,000 cells per sample were used for analysis.

## NanoBiT-G-protein dissociation assay

The β$_2$AR-induced Gs heterotrimer dissociation was measured by a NanoBiT-G-protein dissociation assay[44], in which interaction between a Gα subunit and a Gβγ subunit was monitored by a NanoLuc-based enzyme complementation system called NanoBiT (Promega). Specifically, a NanoBiT-Gs protein consisting of Gα$_s$ subunit fused with a large fragment (LgBiT) at the alpha-helical domain, and an N-terminally small fragment (SmBiT)-fused Gγ$_2$ subunit with a C68S mutation was expressed along with untagged Gβ$_1$ subunit in HEK293 cells deficient for the β$_2$AR and β-arrestin1/2[42]. The cells were passaged and seeded in a 6 cm culture dish at a concentration of $2 \times 10^5$ cells ml$^{-1}$ with 4 mL of DMEM (Nissui) supplemented with 10% fetal bovine serum (Gibco), glutamine, penicillin, and streptomycin 1-day before transfection. Transfection solution was prepared by combining 8 μL (per dish hereafter) of polyethyleneimine (PEI) Max solution (1 mg ml$^{-1}$; Polysciences), 400 μL of Opti-MEM (Thermo Fisher Scientific), and a plasmid mixture consisting of 200 ng LgBiT-containing Gα$_s$ subunit, 1 μg

Gβ$_1$, 1 μg SmBiT-fused Gγ$_2$ (C68S), 200 ng RIC8B and 400 ng (or 40 ng or 80 ng) test β$_2$AR construct. After incubation for 1 day, transfected cells were harvested with 0.5 mM EDTA-containing Dulbecco's PBS, centrifuged, and suspended in 4 mL of HBSS containing 0.01% bovine serum albumin (BSA; fatty acid−free grade; SERVA) and 5 mM HEPES (pH 7.4) (assay buffer). The cell suspension was dispensed in a white 96-well plate at a volume of 80 μL per well and loaded with 20 μL of 50 μM coelenterazine (Carbosynth) diluted in the assay buffer. After 2 h incubation at room temperature, the plate was measured for baseline luminescence (SpectraMaxL with SoftMax Pro software 7.03, Molecular Devices), and a titrated test ligand (isoproterenol, epinephrine, or clenbuterol; 20 μL of intermediate serial dilution prepared at a 6x final concentration) were manually added. The plate was immediately read at room temperature for the following 5 min at a measurement interval of 20 sec with an accumulation time of 0.17 s per reading. The luminescence counts over 3−5 min after ligand addition were averaged and normalized to the initial count. The fold-change values were further normalized to that of vehicle-treated samples and were used to plot the G-protein dissociation response. Using Prism 8 software (GraphPad Prism), the G-protein dissociation signals were fitted to a four-parameter sigmoidal concentration-response curve, from which $pEC_{50}$ values (negative logarithmic values of $EC_{50}$ values) and relative $E_{max}$ values were used to calculate mean and SEM. A change in $pEC_{50}$ and relative $E_{max}$ values from that of WT ($\Delta pEC_{50}$ and relative $E_{max}$, respectively) were calculated for each experiment and used for statistical analyses.

## GTP turnover assay

The GTP turnover experiments were conducted by using a modified protocol of the GTPase-Glo assay (Promega) as previously reported[15]. To evaluate the effect of CT on receptor activation, the full-length β$_2$AR-s-CT with the sortase cleavage site was expressed and purified with MNG detergent (NG-310, Affymetrix Anatrace). The purified β$_2$AR-s-CT was diluted to 10 μM and preincubated without/with 5 μM SrtA at 22°C for 3 h in the presence of selected ligands before GTP turnover assay in a buffer condition (20 mM HEPES, pH 7.5, 100 mM NaCl, 0.01% MNG, 0.001%CHS, 1 mM DTT, 10 mM CaCl2). The cleavage condition was ended by adding 10 mM EDTA for 20 min on ice, and SrtA was added to balance all reactions. Subsequently, the GTP turnover was initiated by mixing an equal amount of receptor and Gs to a final concentration of 0.5 μM in 10 μL buffer (20 mM HEPES, pH 7.5, 100 mM NaCl, 0.01% MNG, 0.001%CHS, 100 μM TCEP, 10 mM MgCl2, 5 μM GTP). During the incubation, the 500x GTP-Glo Reagent (Promega) stock was diluted into water and 10 μM of ADP was added. After incubation for a given period (incubation for 20 min for the agonist condition, or 3 h for the apo condition), 10 μL reconstituted GTPase-Glo reagent was added to the mixture to convert the remaining GTP to ATP. After another 30 min at room temperature, 20 μL detection reagent was added to the final mixture and incubated for 10 min before measurement. Luminescence was measured by an EnSight$^{TM}$ multi-mode plate reader (PerkinElmer). The luminescence value of all conditions was normalized by the values obtained with Gs alone. To directly compare ligands efficacy, the GTP turnover of all ligands was compared with the values obtained for Gs alone and normalized to the maximum response in the presence of epinephrine at the 20 min incubation time.

## Expression, purification, and labeling of the β$_2$AR receptor

The expression and purification of the β$_2$AR receptor are according to methods described previously[26]. Briefly, the β$_2$AR receptor with an N-terminal FLAG tag was expressed in *Sf*9 insect cell cultures for 48 h after infection with recombinant baculovirus (BestBac, Expression Systems). Cells were lysed, and the β$_2$AR was extracted using the detergent *n*-dodecyl-β-D-maltopyranoside (DDM). M1 Flag affinity chromatography was used as the initial purification step, followed by

alprenolol-sepharose chromatography for the selection of functional receptors. After that, the receptor was exchanged to MNG-CHS buffer on the M1 Flag affinity column. The eluted receptor was treated with lambda phosphatase to remove heterogeneous phosphorylation. For the smFRET and PRE experiments, receptors were labeled with fluorophore or spin reagent after de-phosphorylation. Receptors were loaded onto a Superdex 200 column equilibrated in a buffer (20 mM HEPES, pH 7.5, 100 mM NaCl, 0.01%MNG, 0.001%CHS, 100 μM TCEP). Subsequently, the concentrated sample was then aliquoted, flash-frozen, and stored at −80 °C in the presence of 25% glycerol.

For site-specific labeling before smFRET experiments, the receptor was diluted to 10 μM in buffer (20 mM Hepps pH 7.5, 100 mM NaCl, 0.01%MNG, 0.001%CHS, 100 μM TCEP, 30 μM Atenolol). A 3-fold molar excess of maleimide Cy5 (Cytiva) and 6-fold of maleimide Alexa Fluor 555 (Thermo Fisher Scientific) were incubated with the receptor at room temperature for 30 min. Excess dyes were quenched with 10 mM Cysteine and then removed using a G50 desalting column. The concentrated sample was run on a Superdex 200 column equilibrated in a buffer (20 mM HEPES, pH 7.5, 100 mM NaCl, 0.01% MNG, 0.001% CHS). The fractions containing homogeneous receptors were pooled, concentrated, aliquoted, flash-frozen, and stored at −80°C in the presence of 25% glycerol.

For NMR and PRE experiments, the GST-CT fusion protein in the pGEX 6p-1 vector was transformed into *E. coli* cells BL21(DE3) (CWBIO), and cells were collected when the $OD_{600}$ reached 1.0 in the Terrific Broth medium. For the isotopic backbone labeling experiment, cells were centrifuge, collected, and resuspended in M9 minimal medium, supplemented with $^{15}$N labeled ammonium chloride (Cambridge Isotope Laboratories) or U–$^{13}$C6 labeled glucose (Cambridge Isotope Laboratories) as the nitrogen and carbon sources, respectively. The GST-CT fusion protein was purified using Ni-NTA chromatography, followed by the addition of TEV protease to remove the GST at room temperature for 3 h. Then, the sample was boiled at 60 °C for 30 min to precipitate GST and TEV protease, and the soluble CT was further purified on a Superdex 75 column equilibrated with buffer (20 mM HEPES, pH 7.5, 100 mM NaCl). The CT was concentrated for fusion reactions.

The purification of SrtA and optimization of the ligation reaction followed a previously reported protocol[66]. Briefly, the sortase reactions were conducted in a buffer (20 mM Hepes, pH 7.5, 100 mM NaCl, 0.01% MNG, 0.001% CHS, 5 mM β-mercaptoethanol (β-ME), 30 μM Atenolol, 2.5 μg/ml leupeptin, 160 μg/ml benzamidine and 10 mM CaCl2). The receptor at 5 μM, CT at 50 μM, and SrtA at 1 μM were incubated at 4 °C overnight. The ligated mixture was loaded to Ni-NTA chromatography and followed by M1 Flag affinity chromatography to remove the extra CT and SrtA. The ligated receptor was treated with thrombin protease to remove C-terminal His tag, and loaded onto a Superdex 200 column equilibrated in buffer (20 mM HEPES, pH 7.5, 100 mM NaCl, 0.01%MNG, 0.001%CHS), and the eluted sample was concentrated, flash-frozen for further study.

## Expression and purification of the Gs protein and Nb6B9
The expression and purification of Gs protein followed a previously reported protocol[26]. The human $G\alpha_s$, $His_6$-rat $G\beta_1$, and bovine $G\gamma_2$ were co-expressed in *Trichoplusia ni* insect cells grown in ESF 921 serum-free medium (Expression Systems). Cells were infected with the baculoviruses at a density of $3 \times 10^6$ cells ml$^{-1}$ followed by an incubation of 48 h at 27 °C. Cells were harvested by centrifugation and lysed in a buffer comprised of 10 mM Tris, pH 7.5, 100 μM MgCl2, 5 mM β-ME, 30 μM GDP, 2.5 μg/ml leupeptin, and 160 μg/ml benzamidine. Cell membranes were prepared and solubilized in buffer (20 mM HEPES, pH 7.5, 100 mM NaCl, 1% sodium cholate, 0.05% MNG, 5 mM MgCl2, 0.5 μl CIP for 100 ml buffer, 5 mM β-ME, 10 μM GDP, 2.5 μg/ml leupeptin, 160 μg/ml benzamidine). After the removal of insoluble debris, the supernatant was loaded onto Ni resin and washed with buffer

without sodium cholate. Gs protein was eluted and de-phosphorylated by adding lambda phosphatase, CIP, and Antarctic phosphatase on ice for 30 min. 3 C protease was added to remove his tag on $G\beta_1$. The mixture of Gs protein and 3 C protease was dialyzed against the elution buffer at 4 °C overnight to remove imidazole. The dialyzed mixture was reloaded on Ni resin, the flow-through was collected, and salt concentration was adjusted to around 80 mM before loading onto the MonoQ column (GE Healthcare). The elution from MonoQ was concentrated and injected into a Superdex 200 column (GE Healthcare) equilibrated in a buffer (20 mM HEPES, pH 7.5, 2 mM MgCl2, 100 mM NaCl, 0.01%MNG, 0.001%CHS, 100 μM GDP, 100 μM TCEP). For Gs$^{GDP}$ related experiments in smFRET studies, we removed the free GDP on Superdex 200 equilibrated with a buffer without GDP. The Gs protein was concentrated, aliquoted, flash-frozen, and stored at −80 °C.

Nb6B9, containing an N-terminal signal sequence and a C-terminal 8× histidine tag in the pET26b vector, was transformed into *E. coli* cells BL21(DE3) (CWBIO). Cells were induced in Terrific Broth medium with 1 mM IPTG at $OD_{600}$ of 1.2 and cultured with shaking at 22 °C for 20 h. Periplasmic protein was obtained by osmotic shock, and the nanobodies were purified using Ni-NTA chromatography, followed by a Superdex 200 column equilibrated in buffer (20 mM HEPES, pH 7.5, 100 mM NaCl). The eluted sample was concentrated, aliquoted, flash-frozen, and stored at −80°C.

## Single-molecule FRET experiments on a total internal reflection fluorescence (TIRF) microscope
Single-molecule FRET measurements were performed on a home-built objective-type TIRF microscope, based on a Nikon Eclipse Ti-E with an EMCCD camera (Andor iXon Ultra 897), and solid-state 532 and 640 nm excitation lasers (Coherent Inc. OBIS Smart Lasers) as previously described[67]. All smFRET movies were collected using Cell Vision software (Beijing Coolight Technology) and performed at 25 °C in imaging buffer (50 mM HEPES, pH 7.5, 100 mM NaCl, 5 mM MgCl2, 5 mM CaCl2, 0.01% MNG, 0.001%CHS) with an oxygen-scavenging system containing 3 mg/mL glucose, 100 μg/mL glucose oxidase (Sigma-Aldrich), 40 μg/mL catalase (Roche), 1 mM cyclooctatetraene (COT, Sigma-Aldrich), 1 mM 4-nitrobenzylalcohol (NBA, Sigma-Aldrich), 1.5 mM 6-hydroxy-2,5,7,8-tetramethyl-chromane-2-carboxylic acid (Trolox, Sigma-Aldrich). All buffer exchange during concentration titration experiments was directly performed in slide chambers, and buffer with imaging components at 10-fold excess of chamber volume was used for buffer exchange and incubated for 5 min at 25 °C before smFRET measurement.

The polyethylene glycol (PEG)-passivated slide was prepared and stored under −20 °C according to a reported procedure[68]. Before smFRET measurements, the PEG-passivated slide was preincubated with streptavidin and biotinylated M1 Fab fragment sequentially, and the N-terminal FLAG-tagged AF555-Cy5 labeled $\beta_2$AR was preincubated with saturating concentration of ligands for 10 min at RT. For all conditions with isoproterenol, 200 μM isoproterenol was used in the smFRET experiment unless specified. Diluted receptors were immobilized to the slide surface through biotinylated M1 Fab. The density of labeled $\beta_2$AR molecules on the surface (typically 2500 donor spots) was optimized to maximize data-collecting efficiency. To evaluate the effect of saturated Nb6B9, 1 μM Nb6B9 was incubated with immobilized isoproterenol-occupied receptor for 5 min in an imaging buffer before measurement. During titration experiments, Nb6B9 at indicated concentrations (from 316 pM to 100 nM) was present in the imaging buffer. To evaluate the effect of sodium chloride concentration on the FRET histogram, the unliganded receptor was initially immobilized in low sodium chloride concentration (50 mM) or high sodium chloride concentration (1 M), then gradually exchanged to higher or lower sodium chloride, respectively. To evaluate the effect of Gs$^{GDP}$, Gs$^{GDP}$ at indicated concentrations (from 3.16 nM to 10 μM) were injected into the chamber in the presence of an imaging buffer. To

evaluate the effect of nucleotide-free agonist-receptor-Gs complex, isoproterenol-occupied receptors were incubated with a 2-fold excess of Gs at 25 °C for 1 h, followed by treatment with apyrase enzyme for another 90 min, then the stable complex was diluted and immobilized for smFRET imaging. For the titration of nucleotides, GDP or GTP at indicated concentration was added to 0.5 μM Gs$^{GDP}$. For the titration of Gs$^{GDP}$ in the presence of an allosteric modulator, 100 μM isoproterenol and 20 μM cmpd-6 or cmpd-15 were presented in the imaging buffer together with the indicated concentration of Gs$^{GDP}$.

All smFRET data were recorded at 100 ms time resolution (10 frames s$^{-1}$) using Cell Vision software (Beijing Coolight Technology). Collected movies were analyzed by a custom-made software program developed as an ImageJ plugin[69]. Fluorescence spots were fitted by a 2-D Gaussian function. The alignment of the donor and acceptor spots was calculated with a variant of the Hough transform. The background-subtracted total volume of the 2-D Gaussian peak was used as raw fluorescence intensity I. FRET efficiency is calculated as $I_A/(I_A + I_D)$, where $I_A$ and $I_D$ are the Cy5 acceptor and AF555 donor fluorescence intensity, respectively. FRET histograms were generated using FRET efficiencies of every frame at the binning step of 0.01 and further normalized by the total number of frames. Usually, hundreds to thousands of individual single-molecule traces were used in our analysis. FRET traces displayed anti-correlation behaviors between donor and acceptor fluorescent signals were picked and further analyzed by a Hidden Markov Model-based software HaMMy[53] to identify different FRET states and to extract dwell times on each state. For the analysis of low FRET and high FRET occupancy, we fitted the overall histogram with two Gaussian peaks. In most cases, the peak center and width are unconstrained during fitting, except for the cases when in Gs concentration is lower than 10 nM or in the absence of Gs, where we fixed the low FRET peak center at 0.38.

### Single-molecule FRET experiments on a confocal microscope

Measurements were performed on a home-built confocal microscope based on a Zeiss AXIO Observer D1 fluorescence microscope with an oil-immersion objective (Zeiss, 100×, NA = 1.4), and solid-state 532 nm excitation laser (Coherent Inc. OBIS Smart Lasers). Laser power at the samples was ~ 50 μW. The sample was dropped on a cleaned PEG-passivated glass coverslip and the laser confocal point was set to ~ 10 μm above the coverslip. Fluorescence signals from the sample passed through a pinhole (diameter 50 μm) and were spectrally separated by interference dichroic (T635lpxr, Chroma), which were further filtered by bandpass filters ET585/65 m (for AF555, Chroma) and ET700/75 m (for Cy5, Chroma) before detected by two APDs (Excelitas, SPCM-AQRH-14). Photon numbers of different detection channels were binned and recorded simultaneously every 1000 μs. For each experimental condition, three or more identical replicates were performed. Each replicate was collected for five minutes. To be classified as a burst, the total photon counts ($f_{AF555} + f_{Cy5}$) in the burst had to be at least 40. Apparent FRET efficiency of each burst was calculated via ($f_{Cy5}$- $b_{Cy5}$)/($f_{Cy5}$- $b_{Cy5}$+$f_{AF555}$- $b_{AF555}$), in which $b_{AF555}$ and $b_{Cy5}$ were background signals of AF555 and Cy5 respectively. $b_{Cy3}$ and $b_{Cy5}$ were measured as average intensities in the AF555 and Cy5 detection channels, respectively, in the absence of labeled $β_2$AR.

### Solution NMR experiment

NMR spectra were obtained at 20 °C on Bruker Avance 500, 700, and 800 MHz spectrometers, all equipped with four RF channels and a triple-resonance cryo-probe with pulsed-field gradients. 2D $^{15}$N-edited heteronuclear single-quantum coherence (HSQC) spectroscopy, three-dimensional HNCA, HNCO, CBCA(CO)NH, and HNCACB experiments were performed to assign the backbone chemical shift of CT. All NMR spectra were processed using NMRPipe[70] and analyzed using

NMRViewJ[71]. The initial NMR experiment of $^{15}$N-labeled [CT] was conducted to evaluate the peak distribution in the 2D $^1$H-$^{15}$N HSQC spectrum. To mimic the conformational constraint of receptor domain to CT behavior, we ligated unlabeled MBP protein to $^{15}$N-labeled [CT]. Because the 2D $^1$H- $^{15}$N HSQC spectra of MBP-[CT] and $β_2$AR-[CT] are similar, we used MBP-[CT] sample for 3D NMR experiments to obtain chemical shift assignment. All chemical shift assignment measurements were performed using MBP-[CT, His] in a buffer (20 mM HEPES, pH 7.2, 100 mM NaCl, 0.1% DDM, 0.01% CHS, 2 mM EDTA, protease inhibitor cocktail (Roche), 10% D$_2$O, 0.1 μM DSS).

Though the initial assignment was done in DDM detergent, we obtained the HSQC spectrum of MBP-[CT, His], MBP-[CT], or $β_2$AR-[CT] in both DDM and MNG conditions, in which the overall spectral changes were similar. To maintain a better thermostability of the receptor and the complex, we conducted all receptor-related experiments in MNG buffer (20 mM HEPES, pH 7.2, 100 mM NaCl, 0.01% MNG, 0.001% CHS, 5 mM MgCl$_2$, protease inhibitor cocktail (Roche), 10% D$_2$O, 0.1 μM DSS).

For agonist-bound receptor measurement, $β_2$AR-[CT] was diluted to 25 μM in the presence of 200 μM BI-167107. For the receptor-Gs complex experiment, 25 μM $β_2$AR-[CT] was mixed with 1.2-fold molar ratio excess of Gs, and incubated for 1 h at 25 °C. 0.5 U apyrase was added to the receptor-Gs complex for another 1.5 h before the NMR experiment. The HSQC spectra of the $β_2$AR-[CT] and $β_2$AR-[CT]-Gs complex were recorded for 2 sequential eight hours. Intensity ratio and chemical shift changes were analyzed with NMRViewJ software. The weighted average $^1$H–$^{15}$N chemical shift changes were calculated using a previously reported formula $Δδ_{av} = ((Δδ_H)^2 + (Δδ_N/5)^2)^{1/2}$.

### Bimane assay

The bimane labeling and fluorescence spectroscopy were performed as previously described[72]. Briefly, the modified $β_2$AR was labeled at room temperature for 30 min in the presence of a 10-fold excess fluorophore. Excess dyes were quenched with 10 mM cysteine and then loaded onto a Superdex 200 column equilibrated in a buffer (20 mM HEPES, pH 7.5, 100 mM NaCl, 0.01%MNG, 0.001%CHS). The GST-CT with native C378 for bimane labeling was expressed in E. coli cells, purified and the N-terminal GST was removed before bimane labeling. Bimane labeled CT was loaded to Superdex 200 column to remove free labeling reagent. Fluorescence spectroscopy experiments were performed on a Duel-FL Fluorometer (Horiba) with the emission scan mode. The excitation was set at 381 nm, and emission was measured from 420–550 nm with an integration time of 0.1 s. For the allosteric effect of multiple factors on the C378 bimane environment, sodium chloride (gradient from 50 mM to 1 M), Nb6B9 (gradient from 0.2 μM to 5 μM), and cmpd-6 (gradient from 0.2 μM to 100 μM) were incubated with 0.2 μM receptor or CT at 20 °C step by step. Measurements were conducted after 5 min of incubation with stirring. The buffer used for these experiments was 50 mM HEPES, pH 7.5, 50 mM NaCl, 0.005% MNG, 0.0005% CHS, and 2 mM MgCl$_2$. All experiments were performed at 20 °C. Fluorescence intensity was corrected by subtracting the fluorescence curve for buffer alone and normalized by the curve obtained with the unliganded receptor. Data were analyzed with OriginLab 9 software, and curves are shown as the mean of independent triplicates.

### Molecular dynamics (MD) simulations

We build an all-atom model of the $β_2$AR comprising amino acids 26-390. VMD1.9.2[73] was used to preprocess the high-resolution crystal structure of the human $β_2$AR (PDB-ID:2RH1). Missing residues were filled using the coordinates of other previously solved 3D structures of the $β_2$AR, namely PDB-ID: 3P0G (residues 26-29), 3SN6 (residues 231-236), 4AMJ (residues 262-266), 2R4S (residues 343-348). Any co-crystallization atoms different from water closer than 5 Å to the protein or the ligand carazolol were removed. CT residues 349-365

were modeled by homology modeling using the SWISS-MODEL server[74]. The search for sequence homologs only provided one protein template (PDB-ID: 5JI4). CT residues 366-390 were built by ab-initio modeling using QUARK[75]. We made sure construct mutations C77V, M96T, M98T, N148C, C265S, C327S, and C341L in the smFRET study were also present in the model. Internal water molecules not present in the crystal structure were modeled using HOMOLWAT[76]. The CHARMM-GUI builder[77] was used to (a) embed the $\beta_2$AR model into a $150 \times 150 \times 124$ Å$^2$ water box made of explicit water molecules; (b) neutralize the global electrostatic charge; (c) to adjust the ionic strength with 0.15 M NaCl; (d) protonate ASP113 and leave the rest of titratable residues of the protein in their dominant protonation state (pH 7.0), and (e) bridge C106-C191 and C184-C190 by disulfide bonds. The solvated system was first geometry-optimized and then relaxed by applying harmonic positional restraints to all Cα atoms of the protein, which were gradually released throughout the equilibration. The equilibration phase was run in the NPT ensemble (barostat: Berendsen; type: semi-isotropic; time constant: 5 ps; reference pressure: 1.013 bar, thermostat: Berendsen, time constant 1.0 ps, reference temperature 310 K) for 30 ns. Three independent exploratory trajectories were spawned from the last snapshot of the equilibrated system using a random seed. These trajectories were run in the NPT ensemble (barostat: Parrinello-Rahman; type: semi-isotropic; time constant: 5 ps; reference pressure: 1.013 bar, thermostat: Nose-Hoover, time constant 1.0 ps, reference temperature 310 K) for 250 ns each and used to study the initial set-up properties including the size of the box to avoid, for instance, errors with periodic boundary conditions during production simulations. All simulations were run using GROMACS v2018.4[78] in combination with the CHARMM36m force field[79]. Ligand parameters for carazolol were generated with the CHARMM General Force Field[80].

## Adaptive sampling scheme

Production MD simulations were carried out following an adaptive sampling strategy[81], where the individual trajectories are started in "batches" or "epochs", with their starting conformations chosen adaptively using a given heuristic. In particular, we used a so-called explore/exploit scheme[82]. This heuristic intends to sample a target feature (the CYS-CYS distance related to the smFRET measurements, in our case) while simultaneously avoiding oversampling a particularly fit conformation. Effectively, this is implemented by scoring each observed MD simulation frame "$i$" with a probability, $P(frame_i)$, of being chosen as starting point for the next epoch defined as:

$$P(frame_i) \propto W_1{}^*(1 - |D_{CYS-CYS}(target) - D_{CYS-CYS}(frame_i)|) + W_2{}^*1/P_{obs} \quad (1)$$

$P_{obs}$ is defined as the normalized number of times a given conformation (defined as the Cartesian coordinates of the ICL3 and the CT) is observed. $W_1$ and $W_2$ are the relative weights of the exploit and the explore components, respectively. By using $W_1 = W_2$ we balance them equally. This way, between two conformations with equally fit $D_{CYS-CYS}$ values, the least observed of them has a higher chance of being reused as a starting frame.

For the CT-$\beta_2$AR system, the reasons for using such a scheme are as follows:

- This approach allows us to sample the conformational space close to the receptor core and thus obtain meaningful statistics on the contact of the CT with the receptor.
- By starting the next set of simulations (next epoch) close to the receptor core, we avoid sampling the vast conformational space of the disordered CT (including receptor-distant conformations) without forcing/biasing a single simulation.

- What 'close to the receptor core' means here is not an arbitrarily chosen quantity, but an experimentally determined value resulting from the CYS-CYS distance of 44 Å of the high-FRET state.

A total of five epochs, each consisting of ten unbiased MD trajectories, each trajectory 200 ns long (2 µs per epoch, 10 µs in total) were simulated. The data are summarized in Fig. 6, Supplementary Figs. 8 and 9, and Supplementary Tables 2, 3, and 4. The representative MD trajectories of the carazolol-bound $\beta_2$AR can be found in the zenodo dataset associated with this paper.

## MD analysis

Analysis of the MD simulation data was carried out using the Python libraries MDTraj[83] and mdciao[84]. Visual 3D inspection of the trajectories was done using VMD[73]. Analysis Ipython Jupyter Notebooks can be found in the zenodo dataset associated with this paper.

## Paramagnetic relaxation enhancement experiment

The paramagnetic relaxation enhancement experiment utilized site-specific spin labeling and high-resolution heteronuclear nuclear magnetic resonance as previously reported[85]. For spin labeling at C261, the ligated receptor after thrombin protease treatment was incubated with 10 fold excess of 4-(2-Iodoacetamido)-TEMPOL(Sigma) at 4 °C overnight. The excess reagent was quenched with cysteine and then removed by Superdex 200 column. Receptor concentration for the PRE experiment was approximately 115 µM, and a standard 2D $^1$H, $^{15}$N-HSQC pulse sequence was utilized to acquire three sequential 8 h spectra using the freshly TEMPOL-labeled receptor-CT. Then, for the measurement of the diamagnetic state spectrum, 2 mM sodium ascorbate dissolved in NMR buffer was added to the NMR tube to reduce the unpaired electron. Acquisition of the diamagnetic sample using parameters that are identical to those used to obtain the paramagnetic spectrum. For the receptor-Nb6B9 complex, 1.2 fold molar ratio excess of Nb6B9 was added to the spin-labeled receptor. All PRE spectra were processed with NMRPipe software and the intensity of assigned cross peaks was measured with NMRViewJ. The intensity ratio ($I_{ox}/I_{red}$) of cross peaks versus protein sequence was calculated with OriginLab 9 software. The error bar indicated the intensity variation of spectra in three sequential measurements.

## Reporting summary

Further information on research design is available in the Nature Portfolio Reporting Summary linked to this article.

## Data availability

The backbone $^1$H, $^{13}$C, and $^{15}$N chemical shift assignments of MBP-[CT, His] in DDM data generated in this study have been deposited in the BMRB database under accession code 51648. The backbone 1H and 15 N assignments of $\beta_2$AR-[CT] and $\beta_2$AR-[CT] + Gs in MNG data generated in this study have been deposited in the BMRB database under accession code 51653 and 51656, respectively. Source data are provided as a Source Data file. Further information and requests for reagents should be directed to and will be fulfilled by the Lead Contact, Brian K. Kobilka (kobilka@stanford.edu) Source data are provided with this paper.

## Code availability

Custom codes used to analyze the smFRET data will be provided upon request.

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

## Acknowledgements

We thank Kayo Sato, Shigeko Nakano, Yuko Sugamura, and Ayumi Inoue (Tohoku University) for their assistance with plasmid preparation, maintenance of cell culture, and the cell-based GPCR assays; the members at the Inoue lab for their assistance with LgBiT purification. We thank the North-German Supercomputing Alliance (HLRN, bec00196) for the allocation of computing time in the Emmy cluster. We acknowledge access to Piz Daint at the Swiss National Supercomputing Centre, Switzerland under the PSI's share with the project psi04. All NMR experiments were performed at the Beijing NMR Center and the NMR facility of the National Center for Protein Sciences at Peking University.

This study was supported by The Beijing Advanced Innovation Center for Structural Biology, Beijing Frontier Research Center for Biological Structure, Tsinghua-Peking Joint Center for Life Sciences, School of Medicine, Tsinghua University (to J.H., J.W.Z. and C.L.C.); The China Postdoctoral Science Foundation Grant (2015M581075 to J.H.); The National Natural Science Foundation of China (21922704, 22277063 and 22061160466 to C.L.C.); The Swiss National Science Foundation (192780, to R.G.G.); The Deutsche Forschungsgemeinschaft (German Research Foundation) through SFB1423, project 421152132, subproject C01 and HI1502/1-2, project 168703014 (to P.W.H); The Stiftung Charitè; and the Einstein Center Digital Future (to P.W.H); The Japan Agency for Medical Research and Development (19gm5910013, JP19gm0010004, JP20am0101095 and JP22ama121038 to A.I.); The Japan Society for the Promotion of Science (JP21H04791, JP21H05113, JPJSBP120213501 and JPJSBP120218801 to A.I.); The Japan Science and Technology Agency (JPMJMS2023 and JPMJFR215T to A.I.); The Uehara Memorial Foundation (to A.I.); Daiichi Sankyo Foundation of Life Science (to A.I.). B.K.K. is an Einstein BIH visiting fellow and a Chan Zuckerberg Biohub Investigator.

## Author contributions

J.H. prepared the β2AR, Gs protein, Nb6B9, and mutant samples, and performed the smFRET experiment, bimane fluorescent assay, solution NMR experiments, and PRE experiments. A.I. performed the constitutive cAMP Glo-sensor assay, the HiBiT assay, the flow cytometry analysis, and the NanoBiT-G-protein dissociation assay. T.I. purified the LgBiT protein. K.K. helped to set up the NanoBiT assay. X.M. and X.S. helped with receptor purification, and J.Z., S.P., and L.Z. helped with smFRET experiments and confocal FRET measurements. C.L.C. guided and supervised the smFRET experiment and data analysis. J.D.Y.Y. performed the NMR assignment of CT, X.N. and H.L. helped with solution NMR experiments, Y.H. and C.J. guided the NMR measurement and data analysis. R.G.G. and G.P.H. designed and performed the MD simulations and analysis, with guidance from P.W.H. J.H. and B.K.K. designed the project and wrote the manuscript. B.K.K. coordinated all experiments and supervised the overall research. All authors contributed to the editing of the manuscript.

## Competing interests

B.K.K. is a co-founder of and consultant for ConfometRx, Inc. The remaining authors declare no competing interests.
