## [Peer Review File · Nature Communications]

Function and dynamics of the intrinsically disordered carboxyl terminus of β 2 adrenergic receptorREVIEWER COMMENTS

Reviewer #1 (Remarks to the Author):

In this manuscript Jie Heng et al investigated the contribution of the intrinsically disordered C-terminal tail (CT) of β 2AR to the receptor activation and signaling. This is an important topic because CT is unstructured and the general quantitative understanding of the role of this domain is limited. Most of the current knowledge about the CT is from analysis of the role of post-translational modifications within the CT in receptor signaling. For example, phosphorylation sites within CT are substrate for GPCR kinases and bind β -arrestins to regulate signaling or regulate other protein-protein interactions. Here, the authors used smFRET, NMR spectroscopy, Paramagnetic relaxation enhancement experiments, and molecular dynamics simulation to monitor the dynamics of the β 2AR CT. They found that CT allosterically can regulate receptor signaling via electrostatic interaction with the receptor core that compete with the G protein binding. The findings are novel and important with impactful implications and of broad interest. The data is of good quality, and controls are generally thorough and comprehensive. I have a few comments for the authors to address.

1) How does the activity of the two FRET constructs 148C-378C and 148C-406C compare with the WT receptor?

2) The authors interpreted the smaller FWHM value for the 148C-378C construct compared to the 148C-406C as middle of the CT being conformationally more homogeneous than the distal part of CT. However, the FRET peak center for 148C-378C is significantly higher than the peak center for 148C-406C (FRET 0.72 vs. 0.56). Considering the nonlinear dependence of FRET on the distance and the maximum sensitivity of FRET to distance is at FRET = 0.5, one expects a generally wider FRET peak for middle FRET than high FRET, even for similar ensemble of distance changes. The Author's interpretation could still be valid but would require some further proof.

3) Are lipid molecules explicitly modeled in the MD simulations? The cytoplasmic leaflet of the plasma membrane bilayer tends to carry a net negative charge. Could this membrane charge affect the interaction of CT with receptor?

4) Some aspects of the data are consistent with the interpretation that there is rapid dynamics (faster than time resolution of measurement) between many conformations. For example, the continuous shift of FRET peak position in NaCl titration experiments would support this. Also, MD simulations imply that the FRET states are themselves an ensemble of many conformational states. On the other hand, some of the analysis support the assumption that there are defined states (for example the dwell-time analysis). This language of defined long-lived states in the inactive and active conformations of receptor is also

reflected in the discussion. Mentioning that interactions are not exactly defined (as in with defined atomic coordinates) and instead multiple weak and transient interactions constitute each of “FRET states” could be helpful to make a distinction.

5) Related to the previous point, considering the many orders of magnitude difference between the MD and FRET experiments, what is the relationship between the FRET states and peaks and the MD states?

6) Please provide more practical details on the fitting procedure for FRET histograms. Did the authors use a global fit to all conditions with fixed fitting parameters (FRET peak center and width)?

7) please provide more details on the percentage of dynamic vs. static smFRET traces. Also what selection criteria was used for traces that were analyzed by the hidden Markov software for dwell-time analysis. This can be important when interpreting the plateau of dwell-time for example in figure 2f.

8) Kind of related to the previous point, it appears that the largest shift in FRET occupancy is at 100 nM of Gs (Figures 2c and 2d) but dwell-times at this concentration are close to the plateau. Why is that? In a 2-state system one expects the ratio of dwell-times to be similar to the ratio of the two peaks at equilibrium.

9) Regarding cmpnd-15, the authors propose that this modulator stabilize the β 2AR in an inactive state. It is not clear how the referenced data exactly show this. And related to that, in the presence of this compound the FRET histogram shows a rightward shift. Is this a new conformation?

10) Some of the experimental conditions are missing from caption or figure panels. For example, Extended figure 4k concentration of ISO, cmpd-6 or-15 is not mentioned in the caption or in the figure. Figure 2f missing which Iso concentration is used for analysis.

Reviewer #2 (Remarks to the Author):

Heng et al. presented a comprehensive study of the β 2AR C-terminus and its interactions with the intracellular surface of β 2AR. The functional importance of the GPCR C-terminus is investigated in cAMP accumulation assays performed in HEK293 cells for several expressed receptors and for detergent-purified β 2AR in a GTPase-Glo assay, which determined the C-terminus acts as an inhibitor. The dynamic behavior of the β 2AR c-terminus and its relative orientation with respect to the receptor “core” is investigated using an array of biophysical techniques including smFRET, solution NMR spectroscopy, and molecular dynamics simulations. A final schematic figure is shown that summarizes the findings from the paper and suggests a model for how the C-terminus interacts with the receptor core and how this interaction is impacted by ligands and the formation of ternary signaling complexes.

This is a challenging topic that has been understudied by the receptor community. More information on the function and dynamics of flexible segments of receptors is needed, and the presented study should be of great interest to several research communities. The authors’ multi-disciplinary approach to the problem is appreciated and key to providing insight into this topic. The presented data are an interesting and insightful addition to the small but growing body of experimental data on receptor flexible regions. Overall I therefore support publication of the presented work once the following comments and questions have been addressed.

The cAMP accumulation assay data are interesting. The presented data comparing the signaling activity of A2AAR with and without its lengthy C-terminus appear to contrast earlier published work reporting the A2AAR C-terminus was essential for signaling. Please see work by Prof. Anne Robinson’s group, Jain et al. *Biomedicines* 2020. How do the authors explain these very different results? This explanation should also be included in the manuscript.

The NMR assignments obtained for MBP-CT and extended to β 2AR-CT and β 2AR-CT/GS should be deposited in an appropriate resource such as the BMRB before the paper is made available online and in print.

How were the chemical shift differences shown in Figure 4b calculated? These appear to contain information about both the ^1H and ^{15}N chemical shifts but this is not clear. Please specify in the corresponding figure legend and methods section

The PRE data in Figure 6c are interesting, but it is not clear to me they are consistent with the model shown in Figure 7. The model shown in Figure 7 for the “active R” state in the High FRET box shows a section of the C-terminus closely interacting with ICL3. The first several amino acids of the C-terminus adjacent to Helix 8 are shown to be farther away from ICL3. This appears consistent with the FRET data and with the NMR chemical shift difference analysis indicating the middle of the β 2AR-CT interacts with the cytoplasmic surface of the receptor. However this does not appear to be reflected by the PRE data. The PRE data appear to support a model where residues in positions 350 to \sim 395 are closer to the spin

label in ICL3 and residues after position 395 are farther from the spin label. The model shown in Figure 7 however places residues in positions 350 to ~375 farther from ICL3 and spin label and residues occurring later in the sequence closer to the spin label. Based on the model in Figure 7, I would expect some of the largest PRE effects to be for residues at the end of the C-terminus. I would also expect to see a significant change in the PRE for residues at the end of the C-terminus upon addition of the nanobody and little change to the PRE for residues at the beginning of the C-terminus close to Helix 8. However, this is not what is observed experimentally.

I do not think the PRE data are incorrect. Rather, while the model shown in Figure 7 is clean and clear, it may have oversimplified the story and in so doing appears to conflict with some of the experimental data. Usually I find these kinds of models helpful to visualize a complex process; however, here I feel that the data and story are much more nuanced than what is presented in Figure 7. Part of the problem is also that Figure 7 suggests static conformations of the loops and C-terminus, which is also not consistent with key ideas from the experimental data. I therefore suggest the authors either remove the figure entirely or heavily modify the figure so that it accurately represents all the data and the dynamic nature of the studied interactions.

As a minor comment related to the above, in Figure 7 I assume the orange loop is ICL3, but it is not indicated in the figure or the figure legend. If the authors include a modified form of this figure, it would be helpful to label the loops, helices 5-7, and show the approximate location of the PRE spin label.

Additional minor comments:

1. There appears to be a disconnect between the first paragraph of the section introduced on page 12, "Localizing interactions between the CT and β 2AR core domain" and the text following it. There is a gap in the manuscript and what appears to be a different font or spacing used for the section following the first paragraph in addition to the abrupt transition. Please revise.
2. Page 13, line 18: here I don't think the use of "tumbling" is quite correct. I would suggest replacing this with a term with a phrase such as "may result in faster local dynamics".
3. The writing in the introduction is polished, and the manuscript as a whole is well written. However, after the introduction there are a number of relatively small typographical errors throughout the text that should be corrected, e.g. on page 5 line 6 is missing "in" between expressed and HEK293, line 11 "by monitor" should be "to monitor", sometimes "G protein" is used and sometimes "G-protein" is used, and so on throughout.

Reviewer #3 (Remarks to the Author):

This manuscript reports very thorough and convincing studies of a hitherto largely neglected aspect of the mechanisms of GPCR efficacy. I am most qualified to comment on the MD simulations, so will largely limit my comments to this aspect. However, I note that the quality of the English fluctuates throughout the manuscript, which should be revised carefully by a native speaker. Some examples are:

"assay by monitor" page 5, line 11

"with having long" page 5, line 14

Numerous cases in which the number (singular or plural) of the verb is incorrect.

Some figure annotations in the text are missing (pages 6 and 16)

MD simulations

The MD simulations are well conducted and the sampling should be adequate. I am a little disappointed in the MD conclusions. I am sure that more information exists in the trajectories. Even though adaptive sampling was used, clustering the conformations found in the simulation should give considerable insight. Was this attempted? The paragraph that starts on line 19 of page 15 is unnecessarily vague. What are the "different conformations", approximately how many are there? What is the ratio between non-extended and extended conformations? What are the frequencies of occurrence of, for instance, significant salt bridges?

This discussion should be deepened and provided in more detail.

Reviewer #4 (Remarks to the Author):

The authors present a compelling story that the B2AR C-tail self-associates with the cytoplasmic face of the receptor. And that this relatively weak interaction attenuates basal and G protein coupling. Whereas orthosteric ligands have little effect on the CT equilibrium, cytoplasmic allosteric modulators push the equilibrium as expected based upon their pharmacology. Overall, this paper was a pleasure to read with a thoughtful approach through and through. I recommend the paper for publication with minor revisions including one additional experiment (comment 8: titration of Nb35 into B2AR/ISO/Gs/apyrase).

1. Page 5, line 6. "...GPCRs expressed HEK293 cells..." should read "...expressed in...".
2. Page 2, line 11. The authors should fully define EP3 as the prostaglandin EP3 receptor.
3. In the first Results section and the Discussion, the authors suggest that this CT regulatory phenomenon may be a function of CT length. They even test the basal activation of four additional receptors with "long CT". The authors should discuss these aspects further:
 - a. Can the authors elaborate on the relationship between CT and basal activation?
 - b. Is the 70 AA B2AR CT unusually long? The authors should explicitly state the lengths of the other four receptors tested.
 - c. Using your MD and other structural results, how long does a CT need to be to interact with ICL2/3?
 - d. How conserved is a positively charged ICL2/3?
4. When Nb6B9 is added, the 406 FWHM sharpens whereas 378 appears to broaden (Extended Data 3i,j). Is that correct? May that suggest transient Nb/378 interactions?
5. For a smFRET non-specialist, what does the abrupt fluorescent drop represent? Photobleaching? Could you please indicate that briefly in the text?
6. Page 7-8. The authors note the smFRET temporal resolution is insufficient to detect spontaneous fluctuations. This is interpreted as meaning there's either no transition or the interconversion is too fast to observe. Yet, multiple FRET distributions appear to contain a weak, but non-zero, low FRET signal (e.g. Extended Data 4a,g,h) suggesting there is fast interconversion between 2+ states. Have the authors confirmed this low FRET (~0.3 FRET distribution) population is statistically insignificant in all undiscussed cases?
7. Related to the above question. Is the sampling of smFRET replicates sufficient to be considered at equilibrium? If so, could you estimate the G between the low-FRET and high-FRET populations? This free energy difference could be roughly interpreted in terms of the number of H-bonds and salt-bridges responsible for the interaction. And then compared to the number of interactions predicted from ED1, ED13, ICL3 etc deletion experiments.
8. Page 9, second paragraph. The authors speculate the CT may engage Gas in an extended conformation. Presumably at the Gas/Nb35 interface? Could the authors test this by titrating Nb35 into B2AR/ISO/Gs/apyrase and observing FRET distribution? Extended Data Fig 5 does not indicate the FRET pair – presumably 148/378?
9. Page 11, lines 18-20. The authors state "If the initial interactions between Gs and the B2AR actively initiated CT dissociation, one would not expect the plateau in the dwell time...". Wouldn't it be more accurate to say that one would not expect a non-zero plateau?

10. Fig 3c. Add ligand names to the panel. Color each ligand to match the scheme in Fig 3b.
11. The PRE data is high quality. There are a few missing assignments in the identified binding hotspots. Did you ever return to your assignment spectra with this PRE information in mind to help assign the last few NH resonances?
12. Page 13, line 30. Incorrect jargon. Chemical shifts are defined as the resonance frequency normalized to the magnetic field, not as changes in peak position. Changes in peak position should be referred to as chemical shift changes/perturbations/etc. Please check the rest of the text for similar errors.
13. Page 16, last sentence. I don't understand the point that's being made in this sentence and Fig 6e.
14. Fig 5.
 - a. Instead of R-CT, B2AR-CT would be more informative and internally-consistent with NMR nomenclature.
 - b. The panel legends are a bit cumbersome (e.g. panel e). They are difficult to follow and may not reproduce well in publication.
 - i. Is R necessary for the panel legends in Fig 5b,e?
 - ii. Maybe some panel legends can be streamlined by eliminating the ISO and even cmpd-6 from every line.

We thank the reviewers for their constructive comments. We have addressed all your concerns in the revised manuscript. **Modified or new text is in red font in the revised manuscript.**

Reviewer #1

In this manuscript Jie Heng et al investigated the contribution of the intrinsically disordered C-terminal tail (CT) of β 2AR to the receptor activation and signaling. This is an important topic because CT is unstructured and the general quantitative understanding of the role of this domain is limited. Most of the current knowledge about the CT is from analysis of the role of post-translational modifications within the CT in receptor signaling. For example, phosphorylation sites within CT are substrate for GPCR kinases and bind β -arrestins to regulate signaling or regulate other protein-protein interactions. Here, the authors used smFRET, NMR spectroscopy, Paramagnetic relaxation enhancement experiments, and molecular dynamics simulation to monitor the dynamics of the β 2AR CT. They found that CT allosterically can regulate receptor signaling via electrostatic interaction with the receptor core that compete with the G protein binding. The findings are novel and important with impactful implications and of broad interest. The data is of good quality, and controls are generally thorough and comprehensive. I have a few comments for the authors to address.

Response: We thank the referee for giving positive feedback on this manuscript. We made modifications to the manuscript according to your comments and suggestions.

1) How does the activity of the two FRET construct 148C-378C and 148C-406C compare with the WT receptor?

Response: The minimal cysteine template (min-C- β 2AR) and the labeling site, N148C, have been previously reported (Xiao Jie Yao, *PNAS*, 2009 and G. Glenn Gregorio, *Nature*, 2017). These studies showed that the ligand binding and *in vitro* Gs coupling ability of min-C- β 2AR and the N148C mutation were preserved. C378 and C406 are naturally occurring cysteines. We remark on this on **page 6, line 17**.

2) The authors interpreted the smaller FWHM value for the 148C-378C construct compared to the 148C-406C as middle of the CT being conformationally more homogeneous than the distal part of CT. However, the FRET peak center for 148C-378C is significantly higher than the peak center for 148C-406C (FRET 0.72 vs. 0.56). Considering the nonlinear dependence of FRET on the distance and the maximum sensitivity of FRET to distance is at FRET = 0.5, one expects a generally wider FRET peak for middle FRET than high FRET, even for similar ensemble of distance changes. The Author's interpretation could still be valid but would require some further proof.

Response: In addition to the smaller FWHM, our NMR results and DE mutations directly indicate that 378 is closer to acidic amino acids that may directly interact with the receptor core to stabilize its conformation. PRE results also indicate that the 406 site is far away from the spin labeling site 261C (Figure 6C). Please see the modified text on **page 7, line 3**.

3) Are lipid molecules explicitly modeled in the MD simulations? The cytoplasmic leaflet of the plasma membrane bilayer tends to carry a net negative charge. Could this membrane charge affect the interaction of CT with the receptor?

Response: We agree with the reviewer that membrane charge could certainly affect the interaction of the CT with the receptor in the same fashion it affects the interaction between the G protein and the receptor, as we recently showed in the β 2AR-Gi complex system (M. J. Strohman, *Nature communication*, 2019). Nevertheless, due to the limited time resolution of this technique, our simulations focused on the interaction between the CT and the receptor. Thus, while we explicitly modeled lipid molecules, we just used one prototypical neutral phospholipid species, namely POPC,

so that our modeled membrane had no net charge. Therefore, no meaningful interactions of the leaflet with the CT were expected.

Just to be sure, and following the referee's suggestion, we have checked again and found no significant interactions except for the first five residues (L342-S345) of the CT, which immediately follow H8 at the proximal part of the CT (see fig. below).

4) Some aspects of the data are consistent with the interpretation that there is rapid dynamics (faster than time resolution of measurement) between many conformations. For example, the continuous shift of FRET peak position in NaCl titration experiments would support this. Also, MD simulations imply that the FRET states are themselves an ensemble of many conformational states. On the other hand, some of the analysis support the assumption that there are defined states (for example the dwell-time analysis). This language of defined long-lived states in the inactive and active conformations of the receptor is also reflected in the discussion. Mentioning that interactions are not exactly defined (as in with defined atomic coordinates) and instead multiple weak and transient interactions constitute each of “FRET states” could be helpful to make a distinction.

Response: We agree with the comment and appreciate the suggestion. The relatively long high-FRET dwell times alone can be interpreted as the CT having a stable interaction with a specific site on the cytoplasmic surface. It is possible that the interactions between the CT and the cytoplasmic surface could change without resulting in a noticeable change in the high-FRET value. We include the definition of “multiple weak and transient interactions and long-lived smFRET states” in the the revised manuscript (**Page 15, lines 26-31; Page 16, lines 1-6; Page 17, line 31; Page 18, lines 1-6**).

5) Related to the previous point, considering the many orders of magnitude difference between the MD and FRET experiments, what is the relationship between the FRET states and peaks and the MD states?

Response: The MD trajectories sample a much shorter timescale (nanoseconds to microseconds) than the FRET experiment (milliseconds). Out of these trajectories emerges an ensemble of states that are conformationally different, short-lived, and quickly interconverting with one another. Furthermore, most of these states show non-extended conformations (cf. Extended Fig. 8e), with the CT remaining near the receptor. These states display Cys-Cys distances compatible with the high-FRET value, where the Cys-Cys distance distribution peaked at around 40 Å and very little extends beyond 50 Å (cf. Extended Fig. 8a, the panel on the top right). Hence, the ensemble as a whole can be assumed to be representative of the high-FRET state. We note this relationship on **page 15, line 30 to Page16, line 6**

6) Please provide more practical details on the fitting procedure for FRET histograms. Did the authors use a global fit to all conditions with fixed fitting parameters (FRET peak center and width)?

Response: In most cases, the peak center and width are unconstrained during fitting, except for the cases when Gs concentrations are lower than 10nM or in the absence of Gs, where we fixed the low FRET peak center at 0.38. We have updated this in the method section (**Page 25, line 41**).

7) please provide more details on the percentage of dynamic vs. static smFRET traces. Also what selection criteria was used for traces that were analyzed by the hidden Markov software for dwell-time analysis. This can be important when interpreting the plateau of dwell-time for example in figure 2f.

Response: We used the Hidden Markov Model (Hammy software) to analyze all traces that display anti-correlation, and traces containing one or more transitions identified by Hammy were assigned as dynamic molecules and used for dwell time and transition rate analysis. Usually, dynamic traces are about 10-20% of total FRET traces in the presence of Gs.

8) Kind of related to the previous point, it appears that the largest shift in FRET occupancy is at 100 nM of Gs (Figures 2c and 2d) but dwell-times at this concentration are close to the plateau. Why is that? In a 2-state system one expects the ratio of dwell-times to be similar to the ratio of the two peaks at equilibrium.

Response: As mentioned before, Figures 2c and 2d are the overall FRET distribution and FRET occupancy of all traces. In figure 2f, our dwell time results come from dynamic traces displaying transition behavior. Although the high FRET dwell time is close to the plateau, the low FRET dwell time is not. The estimated high FRET occupancy is determined by the high FRET dwell time/(the high FRET dwell time + the low FRET dwell time), whereas the estimated low FRET occupancy is determined by the low FRET dwell time/(the high FRET dwell time + the low FRET dwell time). As shown in the figure below, the estimated FRET occupancy is similar to the measured FRET occupancy, which supports their consistency in defining IC50.

9) Regarding cmpnd-15, the authors propose that this modulator stabilize the $\beta 2AR$ in an inactive state. It is not clear how the referenced data exactly show this. And related to that, in the presence of this compound the FRET histogram shows a rightward shift. Is this a new conformation?

Response: Cmpd-15 was identified as a negative allosteric modulator. The crystal structure of the β_2 AR bound to carazolol and cmpd-15 shows that cmpd-15 binds in an intracellular cleft and stabilizes the β_2 AR TM6 in an inactive state (current ref. 47, Xiangyu Liu, Nature, 2017). The structure of the β_2 AR bound to carazolol and cmpd-15 (pdb-5x7d) is essentially identical to the structure of the receptor bound to carazolol alone (pdb-2rh1). Consistent with the stabilization of the inactive state, the binding of cmpd-15 to β_2 AR decreases the affinity of the agonist isoproterenol in a competition binding assay and inhibits arrestin recruitment (Xiangyu Liu, Nature, 2017). Therefore, we speculated that stabilization of the inactive state by cmpd-15 led to a rightward shift in the FRET histogram (Extended Data Fig. 4k), enhanced high FRET occupancy and dwell time (Fig. 3e, and Extended Data Fig. 5l), and a decrease in the low-FRET complex peak (Extended Data Fig. 5k).

10) Some of the experimental conditions are missing from caption or figure panels. For example, Extended figure 4k concentration of ISO, cmpd-6 or-15 is not mentioned in the caption or in the figure. Figure 2f missing which Iso concentration is used for analysis.

Response: We have updated the details in the method section or in the figure legend. Please see the modification on **page 25, line 5**, and the figure legend of **Extended Data Fig 4**.

Reviewer #2

Heng et al. presented a comprehensive study of the β 2AR C-terminus and its interactions with the intracellular surface of β 2AR. The functional importance of the GPCR C-terminus is investigated in cAMP accumulation assays performed in HEK293 cells for several expressed receptors and for detergent-purified β 2AR in a GTPase-Glo assay, which determined the C-terminus acts as an inhibitor. The dynamic behavior of the β 2AR c-terminus and its relative orientation with respect to the receptor "core" is investigated using an array of biophysical techniques including smFRET, solution NMR spectroscopy, and molecular dynamics simulations. A final schematic figure is shown that summarizes the findings from the paper and suggests a model for how the C-terminus interacts with the receptor core and how this interaction is impacted by ligands and the formation of ternary signaling complexes.

This is a challenging topic that has been understudied by the receptor community. More information on the function and dynamics of flexible segments of receptors is needed, and the presented study should be of great interest to several research communities. The authors' multi-disciplinary approach to the problem is appreciated and key to providing insight into this topic. The presented data are an interesting and insightful addition to the small but growing body of experimental data on receptor flexible regions. Overall I therefore support publication of the presented work once the following comments and questions have been addressed.

Response: We thank the referee for supporting the publication. We made modifications to the manuscript according to your comments and suggestions.

The cAMP accumulation assay data are interesting. The presented data comparing the signaling activity of A2AAR with and without its lengthy C-terminus appear to contrast earlier published work reporting the A2AAR C-terminus was essential for signaling. Please see work by Prof. Anne Robinson's group, Jain et al. Biomedicines 2020. How do the authors explain these very different results? This explanation should also be included in the manuscript.

Response: In the Biomedicines 2020 manuscript, Jain et al. used an untagged A2A construct and measured receptor expression by western blot (thus at a whole-cell level). Therefore, it's not clear what fraction of the A2A- Δ C is expressed on the cell surface, and what fraction is in the ER or the Golgi. Moreover, in the western blot (Figure 2B in the reference) there appears to be considerable degradation of A2A- Δ C.

In our study, we used the N-terminal HA signal sequence (followed by FLAG and HiBiT tags), which is known to enhance surface expression level and is widely used in the GPCR field. We measured surface expression by adding the LgBiT recombinant protein in the conditioned media. Therefore, we believe that differences between our study and that of Jain et al are likely due to expression levels and localization of the truncated receptors.

The NMR assignments obtained for MBP-CT and extended to β 2AR-CT and β 2AR-CT/GS should be deposited in an appropriate resource such as the BMRB before the paper is made available online and in print.

Response: Thanks to the referee for this remark. The NMR data have been deposited to the BMRB database as suggested. The backbone ^1H , ^{13}C , and ^{15}N chemical shift assignments of MBP-[CT, His] in DDM have been deposited in the BMRB under accession number 51648. The backbone ^1H and ^{15}N assignments of β 2AR-[CT] and β 2AR-[CT] + Gs in MNG have been deposited at BMRB under the accession numbers 51653 and 51656, respectively. We updated this on **Page 29, line 24**.

How were the chemical shift differences shown in Figure 4b calculated? These appear to contain

information about both the 1H and 15N chemical shifts but this is not clear. Please specify in the corresponding figure legend and methods section.

Response: The chemical shift differences were calculated by following a previously reported equation of weighted average ^1H – ^{15}N chemical shift changes, $\Delta\delta_{\text{av}} = ((\Delta\delta\text{H})^2 + (\Delta\delta\text{N}/5)^2)^{1/2}$. We have updated it both in the figure legend and methods section.

The PRE data in Figure 6c are interesting, but it is not clear to me they are consistent with the model shown in Figure 7. The model shown in Figure 7 for the “active R” state in the High FRET box shows a section of the C-terminus closely interacting with ICL3. The first several amino acids of the C-terminus adjacent to Helix 8 are shown to be farther away from ICL3. This appears consistent with the FRET data and with the NMR chemical shift difference analysis indicating the middle of the $\beta 2\text{AR-CT}$ interacts with the cytoplasmic surface of the receptor. However this does not appear to be reflected by the PRE data. The PRE data appear to support a model where residues in positions 350 to ~395 are closer to the spin label in ICL3 and residues after position 395 are farther from the spin label. The model shown in Figure 7 however places residues in positions 350 to ~375 farther from ICL3 and spin label and residues occurring later in the sequence closer to the spin label. Based on the model in Figure 7, I would expect some of the largest PRE effects to be for residues at the end of the C-terminus. I would also expect to see a significant change in the PRE for residues at the end of the C-terminus upon addition of the nanobody and little change to the PRE for residues at the beginning of the C-terminus close to Helix 8. However, this is not what is observed experimentally.

Response: Thanks for the referee’s thoughtful suggestions. The revised manuscript includes a new Figure 7, which uses multiple CTs to indicate that CT will adopt multiple conformations, and to align with PRE data, the proximal and middle regions of CT directly interact with ICL loop regions.

I do not think the PRE data are incorrect. Rather, while the model shown in Figure 7 is clean and clear, it may have oversimplified the story and in so doing appears to conflict with some of the experimental data. Usually I find these kinds of models helpful to visualize a complex process; however, here I feel that the data and story are much more nuanced than what is presented in Figure 7. Part of the problem is also that Figure 7 suggests static conformations of the loops and C-terminus, which is also not consistent with key ideas from the experimental data. I therefore suggest the authors either remove the figure entirely or heavily modify the figure so that it accurately represents all the data and the dynamic nature of the studied interactions.

Response: As noted above we have modified Figure 7, which includes multiple CTs to indicate that CT will adopt multiple conformations. The $\beta 2\text{AR-Gs}$ complex formation disrupts the interaction between CT and ICL loop regions.

As a minor comment related to the above, in Figure 7 I assume the orange loop is ICL3, but it is not indicated in the figure or the figure legend. If the authors include a modified form of this figure, it would be helpful to label the loops, helices 5-7, and show the approximate location of the PRE spin label.

Response: In the revised Figure 7, we labeled loops, helices, and the PRE spin labeling site

Additional minor comments:

1. There appears to be a disconnect between the first paragraph of the section introduced on page 12, “Localizing interactions between the CT and $\beta 2\text{AR}$ core domain” and the text following it. There

is a gap in the manuscript and what appears to be a different font or spacing used for the section following the first paragraph in addition to the abrupt transition. Please revise.

Response: We have revised the font and spacing. Please check the modification on **Page 12, line 27**.

2. Page 13, line 18, here I don't think the use of "tumbling" is quite correct. I would suggest replacing this with a term with a phrase such as "may result in faster local dynamics".

Response: Thanks for the suggestion, we have updated the text with "may result in faster local dynamics" on **page 13, line 26**.

3. The writing in the introduction is polished, and the manuscript as a whole is well-written. However, after the introduction there are a number of relatively small typographical errors throughout the text that should be corrected, e.g. on page 5 line 6 is missing "in" between expressed and HEK293, line 11 "by monitor" should be "to monitor", sometimes "G protein" is used and sometimes "G-protein" is used, and so on throughout.

Response: Thanks for the suggestion, we have updated the text according to the referee's comments. and replace all G Protein with G-protein. Please check the modification on **page 5, lines 6 and 11**.

Reviewer #3

This manuscript reports very thorough and convincing studies of a hitherto largely neglected aspect of the mechanisms of GPCR efficacy.

Response: We thank the referee for commenting positively on this manuscript. We made modifications to the manuscript according to your comments and suggestions.

I am most qualified to comment on the MD simulations, so will largely limit my comments to this aspect. However, I note that the quality of the English fluctuates throughout the manuscript, which should be revised carefully by a native speaker. Some examples are:

"assay by monitor" page 5, line 11

"with having long" page 5, line 14

Numerous cases in which the numer (singular or plural) of the verb is incorrect.

Response: We have updated the text according to your suggestions. We changed the "assay by monitor" to "assay to monitor" and "with having long" to "with long". We have carefully checked and updated the number of the verbs in the manuscript.

Some figure annotations in the text are missing (pages 6 and 16)

Response: We have updated the figure annotations in the manuscript.

MD simulations

The MD simulations are well conducted and the sampling should be adequate. I am a little disappointed in the MD conclusions. I am sure that more information exists in the trajectories. Even though adaptive sampling was used, clustering the conformations found in the simulation should give considerable insight. Was this attempted?

Response: Thanks for the referee's comment. It's true that there is a lot of information in the trajectories, we have expanded the MD discussion (beginning **Page 15, line 26** and the Discussion). The referee brought up a very interesting point. Yes, we indeed clustered the conformations following geometric criteria, specifically; we performed a PCA analysis on the Cartesian coordinates of the CT and ICL3.

The paragraph that starts on line 19 of page 15 is unnecessarily vague. What are the "different conformations", approximately how many are there?

Response: The clustering analysis yielded approximately 30 clusters. Their relative weights were, however, clearly fragmented, without a predominant conformation, i.e. the most populated cluster contained only 11% of the simulated data. In the figure below, we use a violin plot to display, together, both the per-cluster Cys-Cys distance (left y-axis) as well as cluster population (inscribed inside the violin, e.g. 11%, 9%, 7%... from left to right). The clusters are shown in descending order of population and the accumulated population (as more clusters are considered) is shown as a gray solid line, to be read on the right y-axis.

Furthermore, these conformations interconvert quickly within the time scale of our individual trajectories (i.e. 250 ns for each trajectory). To be more precise, over 90% of the trajectories contain two or more clusters, as is shown in the figure below.

Overall, this behavior is consistent with the diffusive energy landscape in an intrinsically disordered region.

What is the ratio between non-extended and extended conformations?

Response: As we describe in the manuscript, our data contain very few extended conformations of the CT: “Notably, in most of these conformations, the CT tends to adopt non-extended conformations and stay near the receptor” on **Page 15, Line 28**. This result is clearly depicted in Extended Fig. 8, top right panel, where values higher than 60 Å are rarely seen within the full distribution of sampled Cys-Cys distances. As shown in Extended Fig. 9, further breakdown into relative populations of different Cys-Cys ranges shows that most distances lie within 20-60 Å, namely:

- 0-20 Å: 5.4% of data
- 20-35 Å: 45.0 % of data
- 35-60 Å: 49.2% of data
- 60-75 Å: 0.4%

To increase the visibility of these percentages, we have now included them in **Extended Data Table S4**.

What are the frequencies of occurrence of, for instance, significant salt bridges? This discussion should be deepened and provided in more detail.

Response: As we describe in the manuscript, due to the intrinsically disordered nature of this region, most of the interactions are non-specific and have a low frequency. We express this concept both in the text: “While no single interaction appears to dominate, Q337^{8.55}-L342CT always tethers the proximal segment of CT to H8” (**Page 16, Line 8**) as well as Fig.6, where the frequency of occurrence of any interaction is represented by the opacity of each line. The table below

demonstrates that even the most frequent salt bridge in the entire dataset has a frequency of just 12 %. We have included the below table in **Extended Data Table S3**.

frequency	contact
0.12	D380@C-term - R260@ICL3
0.12	C378@C-term - R260@ICL3
0.11	L381@C-term - F240@ICL3
0.11	N352@C-term - R63@12.49
0.10	G351@C-term - R63@12.49
0.10	Y354@C-term - F264@6.26
0.10	L376@C-term - K263@6.25
0.10	Y354@C-term - E249@ICL3
0.09	D386@C-term - R260@ICL3
0.09	P382@C-term - R260@ICL3
0.09	Y354@C-term - R63@12.49
0.09	Q363@C-term - F264@6.26
0.09	Y350@C-term - R63@12.49
0.09	E369@C-term - K263@6.25
0.09	G353@C-term - R63@12.49
0.09	Y354@C-term - R253@ICL3
0.09	L376@C-term - S262@6.24
0.09	F387@C-term - R260@ICL3
0.08	Y354@C-term - S262@6.24

Reviewer #4

The authors present a compelling story that the B2AR C-tail self-associates with the cytoplasmic face of the receptor. And that this relatively weak interaction attenuates basal and G protein coupling. Whereas orthosteric ligands have little effect on the CT equilibrium, cytoplasmic allosteric modulators push the equilibrium as expected based upon their pharmacology. Overall, this paper was a pleasure to read with a thoughtful approach through and through. I recommend the paper for publication with minor revisions including one additional experiment (comment 8: titration of Nb35 into B2AR/ISO/Gs/apyrase).

Response: We thank the referee for supporting the publication. We made modifications to the manuscript according to your comments and suggestions.

1. Page 5, line 6. "...GPCRs expressed HEK293 cells..." should read "...expressed in..."

Response: Thanks, we have updated the text by following suggestions. We change the "expressed HEK293" to "expressed in HEK293 cells" on **page 5, line 6**.

2. Page 2, line 11. The authors should fully define EP3 as the prostaglandin EP3 receptor.

Response: Thanks, we updated the EP3 as "prostaglandin EP3 receptor" on **page 3, line 11**.

3. In the first Results section and the Discussion, the authors suggest that this CT regulatory phenomenon may be a function of CT length. They even test the basal activation of four additional receptors with "long CT". The authors should discuss these aspects further:

a. Can the authors elaborate on the relationship between CT and basal activation?

Response: Our results suggested that the engagement of the CT with the cytoplasmic surface electrostatic interactions prevents access to Gs.

b. Is the 70 AA B2AR CT unusually long? The authors should explicitly state the lengths of the other four receptors tested.

Response: The CT of GPCRs is not conserved, and it was reported to have an average length of 50 amino acids (DOI: 10.1021/bi027224+). Although the β 2AR and the other four GPCRs' CT have a longer length than the average value, our MD simulation and PRE result suggest that the proximal and middle region of β 2AR CT (341-390) may be more important in modulating receptor basal activity. We discuss the selection of the other four receptors on **page 5, line 13**. The detailed sequences of the other four GPCRs are listed in **Supplementary information Table S1**.

c. Using your MD and other structural results, how long does a CT need to be to interact with ICL2/3?

Response: It is hard to come to a general conclusion, but in the case of the β 2AR, MD results indicate that the first 50 amino acids of the CT (341-390) contribute most to the interaction. Please check the result in Figure 6b.

d. How conserved is a positively charged ICL2/3?

Response: According to the bioinformatic analysis of hundreds of GPCRs, the ICL loop is rich in positively charged residues, which is in agreement with the "positive inside rule" of membrane proteins. Nevertheless, the ICL sequence of Class A GPCRs is not conserved. (DOI: 10.1016/j.sbi.2014.08.002 and doi:10.1093/protein/gzi004).

4. When Nb6B9 is added, the 406 FWHM sharpens whereas 378 appears to broaden (Extended Data 3i,j). Is that correct? May that suggest transient Nb/378 interactions?

Response: In Fig3.h, we show the peak center and width of 378 and 406 with or without Nb. Based on the nonlinear dependence of FRET on the distance, the middle FRET region (efficiency ~ 0.5) is more sensitive to distance changes. Therefore, we can't conclude that there are transient Nb/378 interactions.

5. For a smFRET non-specialist, what does the abrupt fluorescent drop represent? Photobleaching? Could you please indicate that briefly in the text?

Response: Yes, the sudden fluorescence signal drop is caused by the photobleaching of the fluorophore. We clarify this as the reviewer suggested. Check Figure 2. legend on **Page 37, line 13**

6. Page 7-8. The authors note the smFRET temporal resolution is insufficient to detect spontaneous fluctuations. This is interpreted as meaning there's either no transition or the interconversion is too fast to observe. Yet, multiple FRET distributions appear to contain a weak, but non-zero, low FRET signal (e.g. Extended Data 4a,g,h) suggesting there is fast interconversion between 2+ states. Have the authors confirmed this low FRET (~0.3 FRET distribution) population is statistically insignificant in all undiscussed cases?

Response: There are several possible explanations for this, like fast fluctuation, and non-specific labeling. We indeed see a small population of low FRET state traces, therefore, we think it represents the CT dissociated from the receptor core. The MD results support the possibility that we see some fluctuating states which show long distances between two labeling sites.

7. Related to the above question. Is the sampling of smFRET replicates sufficient to be considered at equilibrium? If so, could you estimate the G between the low-FRET and high-FRET populations? This free energy difference could be roughly interpreted in terms of the number of H-bonds and salt-bridges responsible for the interaction. And then compared to the number of interactions predicted from ED1, ED13, ICL3 etc deletion experiments.

Response: Yes, smFRET is conducted at equilibrium, which can estimate the free energy difference. However, our simulations have only sampled the high-FRET state, as they were specifically designed to do so via adaptive sampling. Not having sampled the low-FRET state, we cannot compute a difference in free energy between both FRET states from our data.

Regarding the energetics of extending the CT in terms of salt bridges and hydrogen-bonds, as we say in the text "While no single interaction appears to dominate, Q337^{8.55}-L342CT always tethers the proximal segment of CT to H8" (**Page 16, Line 8**). We have now included **Extended Data Table S3** to make this point even more explicit by showing that all interactions have overall frequencies around 10% and below. This makes the quantification of the individual contributions far from conclusive.

8. Page 9, second paragraph. The authors speculate the CT may engage Gas in an extended conformation. Presumably at the Gas/Nb35 interface? Could the authors test this by titrating Nb35 into B2AR/ISO/Gs/apyrase and observing FRET distribution? Extended Data Fig 5 does not indicate the FRET pair – presumably 148/378?

Response: The Extended Data Fig. 5 uses the 148/378 FRET pair, which is now stated in its figure legend. We have conducted titration experiments as suggested, which are shown below.

The pre-formed β 2AR/ISO/Gs/apyrase complex was immobilized to the PEG surface, then imaging buffer or imaging buffer with different concentrations of Nb35 was exchanged. The control sample is immobilized β 2AR/ISO/Gs/apyrase complex exchanges into buffer without Nb35, which shows some complex dissociated. Although there is some complex dissociation, we could see that Nb35 stabilizes the receptor-Gs complex in a different FRET state. Overall, this result agrees with the speculation that Nb35 displaces the CT from the surface of Gs.

Figure. The effect of Nb35 concentration on the FRET distribution of the β 2AR/Gs/apyrase complex.

9. Page 11, lines 18-20. The authors state “If the initial interactions between Gs and the B2AR actively initiated CT dissociation, one would not expect the plateau in the dwell time...”. Wouldn't it be more accurate to say that one would not expect a non-zero plateau?

Response: Thanks, we updated the text based on the referee's suggestions, check on **Page 11, line 26.**

10. Fig 3c. Add ligand names to the panel. Color each ligand to match the scheme in Fig 3b.

Response: Thanks for the referee's suggestions, we updated a new version of Figure 3c.

11. The PRE data is high quality. There are a few missing assignments in the identified binding hotspots. Did you ever return to your assignment spectra with this PRE information in mind to help assign the last few NH resonances?

Response: We thank the reviewer for the suggestion. We tried as suggested, however, we were not able to obtain assignments of the last few residues with the help of PRE information.

12. Page 13, line 30. Incorrect jargon. Chemical shifts are defined as the resonance frequency normalized to the magnetic field, not as changes in peak position. Changes in peak position should be referred to as chemical shift changes/perturbations/etc. Please check the rest of the text for similar errors.

Response: We updated according to the reviewer's suggestions, **Page 13, line 14, line 19, line 23, and Page 14, line 5.**

13. Page 16, last sentence. I don't understand the point that's being made in this sentence and Fig 6e.

Response: The sentence “Interestingly, when deletion of ICL3 or ED123 mutations are combined with compd-6, the high-FRET center is in a similar position observed for Nb6B9 or 1M NaCl (Fig. 6e).” only describes the phenomenon that these conditions show similar FRET center. Because the FRET value is less informative than structural information, we can’t conclude that a similar FRET peak position means a similar conformational state of receptor CT.

14. Fig 5.

a. Instead of R-CT, B2AR-CT would be more informative and internally-consistent with NMR nomenclature.

Response: We update R-CT to β 2AR-CT in figure 5 according to the reviewer’s suggestion.

b. The panel legends are a bit cumbersome (e.g. panel e). They are difficult to follow and may not reproduce well in publication.

i. Is R necessary for the panel legends in Fig 5b,e?

Response: We updated the panel legends of Fig 5. b and e according to the reviewer’s suggestions.

ii. Maybe some panel legends can be streamlined by eliminating the ISO and even compd-6 from every line.

Response: We streamlined the panel legends according to the reviewer’s suggestions.

REVIEWERS' COMMENTS

Reviewer #1 (Remarks to the Author):

The authors have addressed my questions in their revised manuscript. This is a compelling work and of and high impact and I look forward to seeing it published.

Reviewer #2 (Remarks to the Author):

The authors have addressed this reviewer's questions and concerns with their revised submission. The manuscript appears to be stronger and is suitable for publication.

Reviewer #3 (Remarks to the Author):

The authors have now extracted (and provided) far more information derived from the MD simulations. My comments have been considered adequately and the manuscript can be accepted.

Reviewer #4 (Remarks to the Author):

The authors have addressed all previous concerns. I recommend the article for publication.